Manuscript prepared for J. Name
with version 2014/09/16 7.15 Copernicus papers of the LaTeX class copernicus.cls.
Date: 3 November 2017

# Changing transport processes in the stratosphere by radiative heating of sulfate aerosols

Ulrike Niemeier[1] and Hauke Schmidt[1]

[1]Max Planck Institute for Meteorology, Bundesstr. 53, 20146 Hamburg, Germany

*Correspondence to:* U. Niemeier (ulrike.niemeier@mpimet.mpg.de)

**Abstract.** The injection of sulfur dioxide ($SO_2$) into the stratosphere to form an artificial stratospheric aerosol layer is discussed as an option for solar radiation management. Sulfate aerosol scatters solar radiation and absorbs infrared radiation, which warms the stratospheric sulfur layer. Simulations with the general circulation model ECHAM5-HAM, including aerosol microphysics, show consequences of this warming, including changes of the quasi-biennial oscillation (QBO) in the tropics. The QBO slows down after an injection of $4\,Tg(S)yr^{-1}$ and completely shuts down after an injection of $8\,Tg(S)yr^{-1}$. Transport of species in the tropics and sub-tropics depends on the phase of the QBO. Consequently, the heated aerosol layer not only impacts the oscillation of the QBO but also the meridional transport of the sulfate aerosols. The stronger the injection, the stronger the heating and the simulated impact on the QBO and equatorial wind systems. With increasing injection rate the velocity of the equatorial jet streams increases, and the less sulfate is transported out of the tropics. This reduces the global distribution of sulfate and decreases the radiative forcing efficiency of the aerosol layer by 10% to 14% compared to simulations with low vertical resolution and without generated QBO. Increasing the height of the injection increases the radiative forcing only for injection rates below $10\,Tg(S)yr^{-1}$ ( 8 - 18%), a much smaller value than the 50% calculated previously. Stronger injection rates at higher levels even result in smaller forcing than the injections at lower levels.

## 1 Introduction

A large natural source of sulfur in the stratosphere is volcanic sulfur dioxide ($SO_2$). It is known from observations that stratospheric sulfate from a volcanic eruption impacts the climate and influences also stratospheric dynamics. For example, winter warming observed in regions of the northern hemisphere after the eruptions of Mt Pinatubo and Mt Krakatoa are assumed to be caused by dynamical

changes in the stratosphere (Robock (2000), Shindell et al. (2004)). Changes to the quasi-biennial circulation (QBO) (Labitzke, 1994) and the polar vortex (e.g. Bittner et al. (2016)) were also observed. Stratospheric sulfate aerosol scatters solar radiation (short wave, SW) and absorbs in the near infrared and infrared (long wave, LW) radiation bands. The scattering causes a cooling of the surface and the absorption a heating of the stratospheric aerosol layer.

The cooling of the earth's surface observed after the emission of volcanic aerosols is considered a natural example for potential effects of the proposed climate engineering (CE) technique of injecting sulfur into the stratosphere (Budyko (1977), Crutzen (2006)). Such surface cooling is intended but numerical CE studies show that the artificial climate under CE would not be the same as a natural one under the same radiative forcing conditions (Schmidt et al., 2012), because, e.g. CE changes the hydrological cycle (Tilmes et al. (2013), Kravitz et al. (2013)) due to different effects on top of the atmosphere (TOA) and surface radiation (Niemeier et al., 2013). An impact of the warming in the stratosphere on stratospheric dynamics was discussed by Aquila et al. (2014) who simulate changes of the quasi-biennial circulation caused by sulfur injection. For an injection of $1.25\,\mathrm{Tg(S)yr^{-1}}$ the westerly phase of the QBO is prolonged in the lower stratosphere, and the oscillation vanishes with the injection of $2.5\,\mathrm{Tg(S)yr^{-1}}$. These changes in the QBO are triggered by two processes: changes in the thermal wind balance and increased residual vertical wind velocity. The phase of the QBO influences transport processes in the tropics (Plumb (1996), Haynes and Shuckburgh (2000)) and extratropics (Punge et al., 2009). The impact of sulfur injections on the QBO should, therefore, also affect transport processes in the stratosphere in addition to the acceleration of the Brewer-Dobson Circulation (BDC) described by Aquila et al. (2014). The main intention of our study is to determine how changes of the transport of sulfate aerosol in the stratosphere are dependent on the state of the QBO and the jets in the tropical stratosphere. We performed simulations with the General Circulation Model ECHAM5 (Roeckner et al., 2003) coupled to an aerosol microphysics model (HAM) (Stier et al., 2005). We attempt to answer the questions if ECHAM5-HAM simulates similar impacts on the QBO as described in Aquila et al. (2014), and which consequences this has for dynamical processes in the stratosphere, the global distribution of sulfate aerosol, and the cooling efficiency of the artificial aerosol layer.

Niemeier and Timmreck (2015) determined the efficiency of sulfur injections depending on injection rate and injection area. They defined a forcing efficiency: the relation of top of the atmosphere (TOA) radiative forcing caused by the sulfate aerosols to the injection rate. They also discussed the impact of the subtropical transport barrier on the efficiency. A stronger confinement resulted in lower efficiency. However, their model could not generate a QBO. Aquila et al. (2014) showed an intensification of the equatorial jet caused by the impact of sulfur injections. One may hence expect that changing jets caused by stratospheric sulfate heating feeds back to the sulfate distribution. To estimate this effect, we discuss the efficiency of the sulfur injection and compare to earlier results (Niemeier and Timmreck, 2015). In this study, the aerosols are mostly injected in the tropics as this

showed the strongest forcing efficiency in our model (Niemeier and Timmreck, 2015). Injections over a wider latitude band, also outside the tropics, reduces the aerosol load in the tropics and, thus, the impact on the QBO.

    This paper is structured as follows: We give a brief general overview of stratospheric dynamics and the QBO (Section 2) and summarize the explanation given by Aquila et al. (2014) of how the

heated sulfate layer impacts the QBO. The model setup and the simulations performed in this work are described in Section 3. The results of the simulations are described in three parts: The implication of sulfur injections on stratospheric dynamics in Section 4, the transport of sulfate in Section 5, and the radiative forcing and the efficiency of the injection in Section 6.

## 2    Stratospheric dynamics and transport — a short overview

**2.1    Circulation in the stratosphere**

Long-living species such as ozone are transported in a global-scale stratospheric transport regime with rising air in the tropics, the 'tropical pipe' (Plumb, 1996), and descending air at the poles. The stratospheric meridional residual circulation is known as Brewer-Dobson Circulation (BDC). The tropical pipe consists of an area of very low horizontal mixing and high zonal wind speed,

the equatorial jets of the QBO. Breaking Rossby and gravity waves drive the BDC and cause a strong seasonal dependency with strong transport towards the winter hemisphere of mid-latitudes. Additionally, breaking planetary waves cause rapid isentropical, quasi-horizontal mixing in the lower stratosphere. This 'surf zone' reaches from the subtropics to high latitudes (Holton et al., 1995) and combines fast meridional transport with the slow BDC (Butchart et al., 2006). This quasi-horizontal

mixing is the main transport branch for the sulfate aerosol in the lower extratropical stratosphere.

    Sharp gradients of potential vorticity at the edges of the surf zone act as a transport barrier: the polar vortex at high latitudes inhibits transport to the poles in winter months, and the equatorial jets of the QBO contribute to the formation of a reservoir for chemical species in the lower tropical stratosphere (Trepte and Hitchman, 1992). The formed barrier is strongest in heights from about 21

km to 28 km (50 to 15 hPa). The strength of the transport barrier depends on the phase of the QBO. These barriers can be seen as 'eddy-transport-barriers' (Mcintyre, 1995). As a consequence, the BDC has two horizontal transport branches, one below and one above the transport barrier. Transport of engineered sulfate out of the tropics occurs mainly in the lower branch of the BDC but for small particles in high level injection scenarios also in the upper branch.

A schematic diagram of the transport pattern in the stratosphere is e.g. given in Haynes and Shuckburgh (2000). Butchart (2014) provides an overview of the stratospheric dynamic processes described above, as well as related references.

## 2.2 QBO and stratospheric dynamical processes

The QBO is formed by alternating westerly and easterly winds with an average period of about 28 months at the equator. The phases of the wind propagate from the upper stratosphere (about $5\,\mathrm{hPa}$) downward into the tropopause region.

Observations and previous studies show that transport processes in the stratosphere depend on the phase of the QBO (Plumb and Bell (1982)). Equatorward motion in the westerly jet and poleward motion in the easterly jet, both a consequence of the Coriolis force, create a so called Secondary Meridional Circulation (SMC) with opposite vertical winds in the tropics and subtropics. Within the tropical pipe the air is in general rising but the SMC intensifies the vertical velocity in easterly QBO shear and weakens it in westerly QBO shear.

The described circulation is accompanied by isentropic mixing. The isentropic transport in the different QBO phases has been analyzed in detail by O'Sullivan and Chen (1996), Shuckburgh et al. (2001), and Punge et al. (2009). Shuckburgh et al. (2001) and Punge et al. (2009) describe for QBO westerlies a narrow region at the Equator where mixing is strongly inhibited. The surf zone reaches far into the tropics in the winter hemisphere, going from $5°$ to the mid-latitudes, and a second surf zone develops between $5°$ and $15°$ in the summer hemisphere because the QBO westerlies allow the penetration of waves through the tropics into the summer hemisphere. This causes mixing from the tropics into the sub-tropics. The waves are damped where winds become easterly, causing enhanced mixing in this area (about $20°$ N and S) (Punge et al., 2009). Within QBO easterlies there is weak mixing in the tropics and sub-tropics (Shuckburgh et al., 2001) bordered by a region with large potential vorticity gradients in the subtropical summer hemisphere which inhibit mixing Punge et al. (2009). QBO winds have an impact on extratropical wave propagation as westerly winds allow in general the propagation of these waves, different to easterly winds.

## 2.3 QBO and radiative heating of sulfate aerosol

Aquila et al. (2014) simulated changes of the oscillation of the QBO caused by injection of sulfur into the stratosphere. By injecting $1.25\,\mathrm{Tg(S)yr^{-1}}$, the oscillation slows down and phases with westerly wind in the lower stratosphere are prolonged. Injecting at higher altitude causes the oscillation to break down and a constant westerly wind develops in the lower stratosphere. They show an acceleration of the BDC in the tropics and mid-latitudes, but only a small impact on the high-latitude branches of the BDC.

Stratospheric sulfate absorbs infra-red radiation which warms the lower stratosphere. This radiative heating has two consequences: a disturbed thermal wind balance and an increased residual vertical velocity $\omega^*$ (Niemeier et al., 2011). Temperature and vertical wind shear are approximately in thermal wind balance (Andrews et al. (1987) and Baldwin et al. (2001), see Eq. 1b for details). Thus, the consequence of the heated aerosol layer is a vertical wind shear causing an additional

westerly component of the zonal wind above the heated aerosol layer resulting in the prolonged phases of westerlies in the lower stratosphere. The increase of $\omega^*$ extends to much higher vertical levels than just the heating of the aerosols (Aquila et al., 2014). This stronger $\omega^*$ causes a westerly momentum forcing from the vertical advection of the zonal wind component ($-\omega^* u_z$) which may over-compensate easterly momentum deposit from gravity wave dissipation (Aquila et al., 2014). Once this strengthening of the upward advection overwhelms the wave mean-flow interaction in the shear layer, which causes the downward part of the QBO, the QBO oscillation slows down. Figuratively, one can imagine a child running upward on a downward moving escalator, which is less successful the faster the escalator is.

## 3 Description model and simulations

### 3.1 Model setup

The simulations for this study were performed with the middle atmosphere version of the general circulation model (GCM) ECHAM5 (Giorgetta et al., 2006), using the spectral truncation at wavenumber 42 (T42) and 90 vertical layers up to $0.01\,\mathrm{hPa}$. The GCM solves prognostic equations for temperature, surface pressure, vorticity, divergence, and phases of water. In this model version with 90 vertical levels the quasi-biennial oscillation (QBO) in the tropical stratosphere is internally generated (Giorgetta et al., 2006).

The aerosol microphysical model HAM (Stier et al., 2005) is interactively coupled to the GCM. HAM calculates the sulfate aerosol formation including nucleation, accumulation, condensation and coagulation, as well as its removal processes by sedimentation and deposition. A simple stratospheric sulfur chemistry is applied above the tropopause (Timmreck, 2001; Hommel et al., 2011). The sulfate is radiatively active for both, SW and LW radiation, and HAM is coupled to the radiation scheme of ECHAM5. The sulfate aerosol influences dynamical processes via temperature changes caused by scattering of solar radiation and absorption of near-infrared and infrared radiation. Within this stratospheric HAM version, apart from the injected $SO_2$, only natural sulfur emissions are taken into account. These simulations use the model setup described in Niemeier et al. (2009) and Niemeier and Timmreck (2015). The sea surface temperature (SST) is set to a climatological values as in Toohey et al. (2011) and does not change due to CE.

Bunzel and Schmidt (2013) compared simulations with low and high vertical resolution (47 and 95 levels) versions of ECHAM6. The Brewer-Dobson Circulation is qualitatively similar and independent of resolution. The high resolution shows 5% less vertical mass flux and 20% increase in age of air at mid-latitudes. Numerical diffusion is reduced when increasing the vertical resolution (see Land et al. (2002) for an applied and Quarteroni et al. (2010) for an theoretical approach), resulting in lower vertical extent of the sulfate layer. Schmidt et al. (2013) show that differences between the atmospheric mean states and trends simulated with two different vertical resolutions of ECHAM

are in general small except for the tropics where the QBO is a dominant feature. Thus, we presume that we can assign differences between responses to sulfate aerosol forcing simulated with different vertical resolutions mostly to the internally generated or not generated QBO and to different strength of vertical numerical diffusion.

## 3.2 Simulations

We estimate the impact of changes of the QBO phase on transport by varying the injection-rate, -height and -area. We inject 4, 6, 8 and $10\,\mathrm{Tg(S)yr^{-1}}$ at heights of $60\,\mathrm{hPa}$ and $30\,\mathrm{hPa}$ (19 km and 24 km). Injection rates of 4 and $8\,\mathrm{Tg(S)yr^{-1}}$ are chosen to study the impact of the heated aerosol layer on the QBO, as well as the feedback of the changing dynamics on the transport of sulfate. The simulation with an injection of $4\,\mathrm{Tg(S)yr^{-1}}$ allows us to build composites of different QBO-phases to get a direct comparison of their impact on transport. Simulated results using injection rates with $8\mathrm{Tg(S)/y}$ and $10\,\mathrm{Tg(S)yr^{-1}}$ are in general similar because the QBO breaks down in both cases. Natural variation is high in the tropics due to the different QBO phases and also at high latitudes due to a very variable polar vortex. Both reduces the statistical significance of the results and requires long simulation periods. Therefore,when discussing dynamical impacts in Section 4 and Section 5 we base this on the long simulations 8Tg60 and 8Tg30 (Table 1), extended over 40 and 30 years, respectively.

The $10\,\mathrm{Tg(S)yr^{-1}}$ simulations (10Tg60, 10Tg30 and 10Tg60lat30) with a length of 10 years (Table 1) have been performed in order to allow a comparison to results in Niemeier and Timmreck (2015). We show their simulation result (Geo10), which was performed with a 39 level version of ECHAM5-HAM, without internally generated QBO. This allows a direct comparison of the impact of the model resolution and resulting different tropical wind profiles on the results (Section 6). Most simulations are performed with injections into one grid box at the equator. We perform also a simulation where we extend the injection area to a band between $30°$ N and $30°$ S (10Tg60lat30) with the same zonal extension and position as the box (Table 1). This reduces the amount of sulfur injected in the tropics and reduces the radiative heating too. However, Niemeier and Timmreck (2015) showed that this strategy intensifies meridional transport and reduces the forcing efficiency.

All anomalies are calculated relative to the control simulation. Without sulfur injections, this simulation generates a QBO with an average period of about 32 months. Thus, all averages over the timeseries contain both QBO phases. Results in Sections 4 and 5 are averaged over the period given in Table 1. Results in Section 6 are averaged over the last three (Geo10) or four years of the simulation.

**Table 1.** Overview of the parameters for the simulations performed with ECHAM5-HAM. The injection rate differs between the simulations, as well as injection area and height. *Box* is one grid box at the equator at $120.9°$ E to $123.75°$ E and equator to $2.8°$ N. The injection area *$30°$ N to $30°$ S* has in longitudinal direction also the width of one grid box. Furthermore, the globally averaged aerosol optical depth (AOD) and the ratio of the sulfate burdens in the tropics ($10°$ N to $10°$ S) to extratropics ($30°$ to $90°$) are given.

| Simulation name | Injection rate | Injection height | Injection area | Number of vert. levels | Simulation duration | AOD | Burden ratio |
|---|---|---|---|---|---|---|---|
| 4Tg60 | $4\,\mathrm{Tg(S)yr}^{-1}$ | $60\,\mathrm{hPa}$ | box | 90 | 50 y | 0.085 | 1.21 |
| 4Tg30 | $4\,\mathrm{Tg(S)yr}^{-1}$ | $30\,\mathrm{hPa}$ | box | 90 | 20 y | 0.089 | 2.62 |
| 8Tg60 | $8\,\mathrm{Tg(S)yr}^{-1}$ | $60\,\mathrm{hPa}$ | box | 90 | 40 y | 0.13 | 1.34 |
| 8Tg30 | $8\,\mathrm{Tg(S)yr}^{-1}$ | $30\,\mathrm{hPa}$ | box | 90 | 30 y | 0.12 | 2.57 |
| 10Tg60 | $10\,\mathrm{Tg(S)yr}^{-1}$ | $60\,\mathrm{hPa}$ | box | 90 | 10 y | 0.151 | 1.31 |
| 10Tg30 | $10\,\mathrm{Tg(S)yr}^{-1}$ | $30\,\mathrm{hPa}$ | box | 90 | 10 y | 0.131 | 2.56 |
| 10Tg60lat30 | $10\,\mathrm{Tg(S)yr}^{-1}$ | $60\,\mathrm{hPa}$ | $30°$ N to $30°$ S | 90 | 10 y | 0.157 | |
| Geo10 | $10\,\mathrm{Tg(S)yr}^{-1}$ | $60\,\mathrm{hPa}$ | box | 39 | 6 y | 0.176 | 1.18 |
| Control | $0\,\mathrm{Tg(S)yr}^{-1}$ | | | 90 | 50 y | | |

## 4 Implication of sulfur injection for stratospheric dynamics

The injection of sulfur into the stratosphere and the resulting heating by the aerosols cause a change of the QBO frequency in our simulations (Fig. 1). The results are similar to Aquila et al. (2014) but in their simulations the QBO is impacted already at lower injection rate. Different to Aquila et al. (2014) we use a full aerosol microphysical model and simulate the evolution of the aerosol with varying particle sizes. Injecting $4\,\mathrm{Tg(S)yr}^{-1}$ at $60\,\mathrm{hPa}$ prolongs the westerly phase in the lower stratosphere (70 to $25\,\mathrm{hPa}$). The downward propagation of the easterly winds above $25\,\mathrm{hPa}$ is delayed by several months. Injecting $8\,\mathrm{Tg(S)yr}^{-1}$ has the consequence of a complete shutdown of the oscillation. In the lower stratosphere, between $50\,\mathrm{hPa}$ and $25\,\mathrm{hPa}$ a layer with constant westerly winds develops, accompanied by a layer of constant easterly winds above. When injecting at $60\,\mathrm{hPa}$ the semi-annual oscillation (SAO) in the upper stratosphere propagates down to 5 to $10\,\mathrm{hPa}$, comparable to the control simulation. Injection of $8\,\mathrm{Tg(S)yr}^{-1}$ at higher level, $30\,\mathrm{hPa}$, intensifies the westerly jet in strength and increases its vertical extension (up to $5\,\mathrm{hPa}$). The easterly jet and the lower limit of the SAO region are shifted upward.

### 4.1 Effects on stratospheric temperature

We discuss the temperature and wind anomalies for simulations with an injection rate of $8\,\mathrm{Tg(S)yr}^{-1}$ (Fig. 2 and Fig. 3): injection at $60\,\mathrm{hPa}$ for southern hemispheric winter (JJA) and northern winter (DJF), and injection at $30\,\mathrm{hPa}$ only for DJF. The anomalies are relative to the mean over the 50-year control simulation which includes all phases of the QBO.

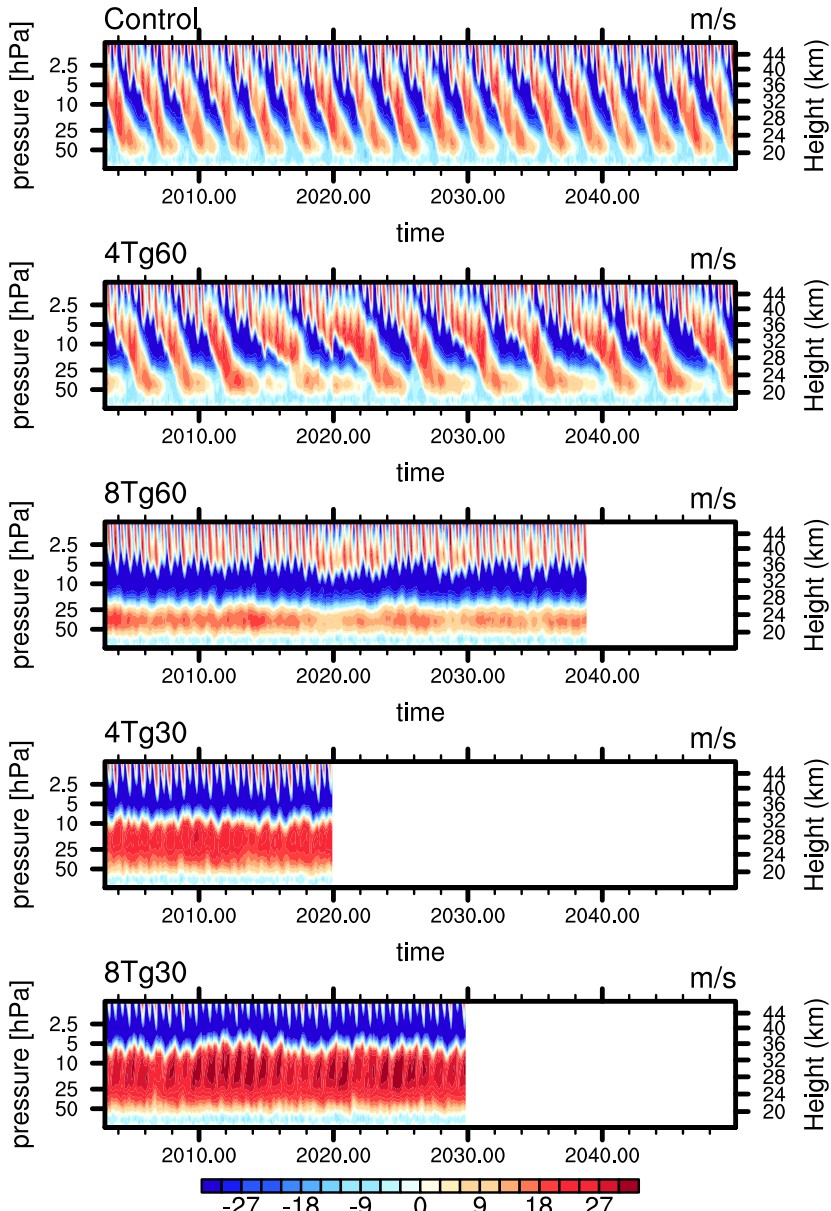

**Figure 1.** Zonal mean zonal wind velocity $\mathrm{ms}^{-1}$ at the equator for the control simulation and simulations with injection rates of $4\,\mathrm{Tg(S)yr}^{-1}$ and $8\,\mathrm{Tg(S)yr}^{-1}$ at a height of $60\,\mathrm{hPa}$ and $30\,\mathrm{hPa}$. The values for height [km] at the right axes are approximations only.

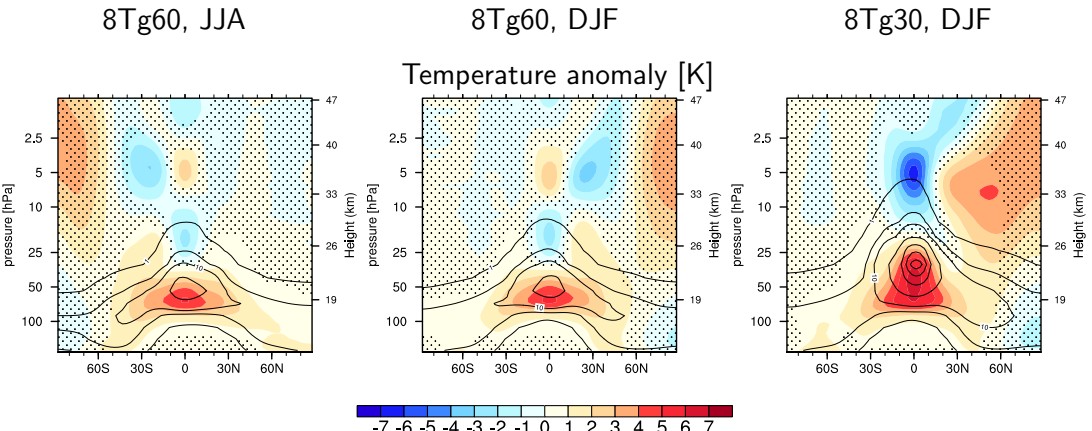

**Figure 2.** Zonal mean temperature anomaly for injection of $8\,\mathrm{Tg(S)yr}^{-1}$ at $60\,\mathrm{hPa}$ (left and middle) relative to control run at northern hemisphere summer (JJA, left) and winter (DJF, right) and DJF for an injection at $30\,\mathrm{hPa}$. Results are compared to a control simulation which includes different phases of the QBO. Stippling indicates areas which are not significant at the 95% level. Contour lines for sulfate aerosol mixing ratio [ppm] are plotted for 1, 5, 10, 25, 50, 75, 100 ppm.

The broad temperature anomaly in the lower stratosphere is caused by the absorption of ~~LW~~ ra-
215  diation through sulfate and, thus, reflects the position of the aerosol layer (Fig. 2). The strongest
warming, between $30°$ N to $30°$ S and $50\,\mathrm{hPa}$ to $100\,\mathrm{hPa}$, occurs just below the maximum sulfate
mixing ratio. The positive anomaly does not extend to the pole in the winter hemispheres. This tem-
perature anomaly in the heated sulfate layer is significant at the 95% level, calculated using a Student
t-test. Above this heated layer the typical temperature pattern with anomalies of opposite sign at the
220  equator and in the subtropics appears, related to the secondary meridional circulation (SMC) of
equatorial jets. In our results theses signals are significant in the winter hemisphere, e.g. JJA at $30°$S
at $20\,\mathrm{hPa}$ and $5\,\mathrm{hPa}$, opposite to the positive anomaly of more than 3 K at the winter pole. Internal
variability around the polar vortex is too high to allow significant results with timeseries of only 30
and 50 years.
225  Injecting at $30\,\mathrm{hPa}$ (8Tg30) results in a vertically more extended aerosol layer. The sedimentation
path is longer and the aerosol is injected into an area were the tropical pipe dominates and merid-
ional transport is lower than in 8Tg60 (see Section 2). Therefore, in 8Tg30 the heating of the sulfate
aerosol extends up to a height of $25\,\mathrm{hPa}$. A consequence is a reduction of the equator to pole tem-
perature gradient in the upper stratosphere, and an increase of the gradient in the lower stratosphere.
230  The temperature anomalies in the upper stratosphere, including the cooling above the heated aerosol

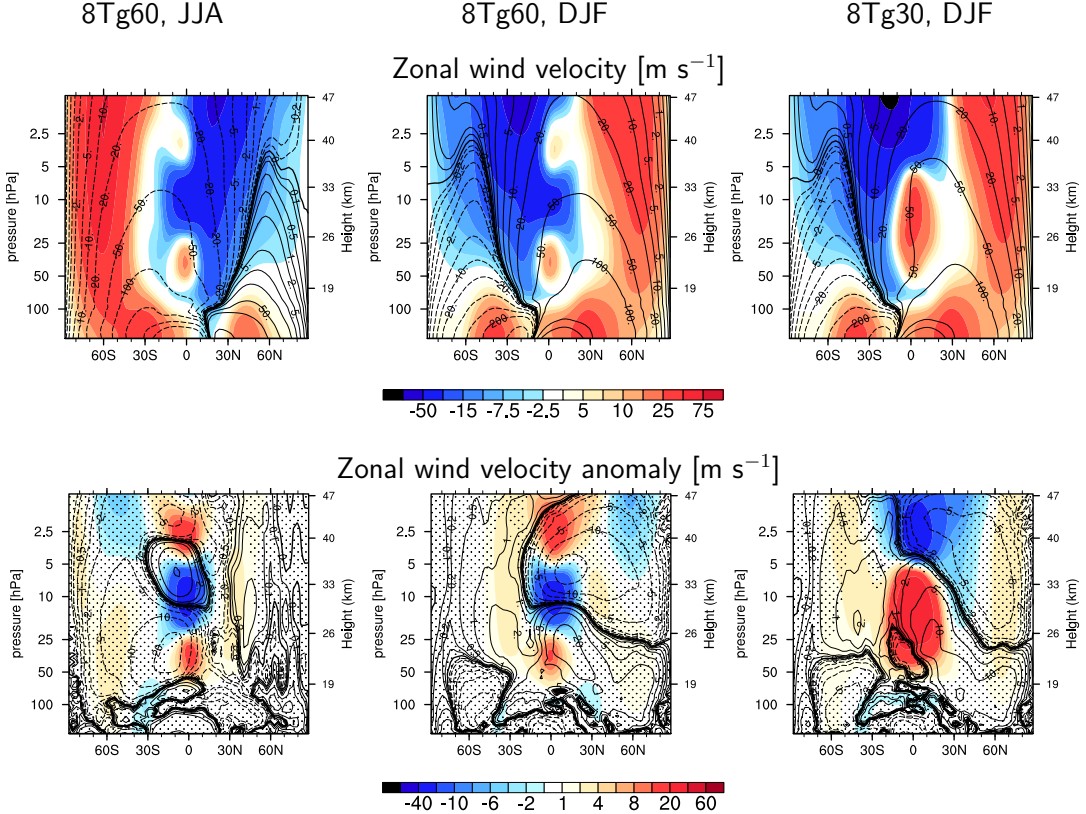

**Figure 3.** Zonal mean zonal wind velocity (top) and the anomaly of the zonal wind velocity (bottom) for injection of $8\,\mathrm{Tg(S)yr^{-1}}$ at $60\,\mathrm{hPa}$ (left and middle) relative to control run at northern hemisphere summer (JJA, left) and winter (DJF, middle) and DJF for an injection at $30\,\mathrm{hPa}$ (right). Contour lines show the stream function $[\mathrm{kg\,s^{-1}}]$ (top) and the difference of the streamfunction to the control simulation (bottom) for the two seasons. Results are compared to a control simulation which includes different phases of the QBO. Stippling indicates areas which are not significant at the 95% level.

layer in the tropics, are caused by the increase of the residual vertical wind, vertical advection, and the related adiabatic heating anomalies (Toohey et al., 2014).

## 4.2 Effects on zonal and meridional wind

A dominating feature of the zonal winds are the polar night jets (Figure 3, top). The velocity in the
235 equatorial jets of the QBO, in Control as well as in 8Tg60, is about a factor of three smaller than in the polar night jet. This changes when injecting at $30\,\mathrm{hPa}$ (8Tg30). Then, the velocity of the equato-

rial jet is comparable to the velocity of the polar jet at a similar altitude. Also the vertical extension of the equatorial jet increases to a height of 5 hPa. The vertical extension of the jet is coupled to the temperature anomaly, with the maximum velocity of the westerly jet just above the heated aerosol layer, and adiabatic cooling by the SMC above. Therefore, the changes in the QBO winds, caused by the heating of the sulfate layer in the lower stratosphere, and related vertical advection anomalies extend the temperature anomalies into the upper and extratropical stratosphere. In 8Tg30 the westerly jet is much wider than in 8Tg60. The westerly component of the wind anomaly is extended into the subtropics, most probably related to better wave propagation within westerly winds, as described in Section 2.2.

In both simulations the summer easterlies are weakened around $30°$, the only significant impact of the sulfate on the zonal wind outside of the tropics. The polar night jet partly intensifies in lower stratosphere of 8Tg60 and intensifies and is pushed poleward in 8Tg30. However, this is not a significant signal.

## 4.3  Effects on the Brewer-Dobson Circulation

The black contours in Fig. 3 (top) show isolines of the mass stream-function of the residual circulation. Positive (solid) streamlines describe clockwise circulation, negative (dashed) ones counterclockwise circulation. The streamlines represent the BDC with the overturning circulation in the winter hemisphere, but they do not show wave induced mixing in the surf-zone and transport barriers (Haynes and Shuckburgh, 2000). The contour lines in Fig. 3 (bottom) show the anomaly of the streamlines with respect to the control simulation. Dashed lines indicate negative anomalies which would be an intensification of counter-clockwise circulation. Solid lines indicate positive anomalies and an intensification of clockwise circulation.

Compared to the control run (anomaly in Fig. 3, bottom) the vertical winds in the tropical pipe are intensified in both simulations at the equator,e.g. up to 10 hPa in 8Tg60. The streamline rises higher in 8Tg30 which indicates the intensified tropical pipe in the westerly jet in 8Tg30. The downward motion at the winter pole intensifies only at the South Pole through the whole stratosphere, not at the North Pole, where we cannot confirm the results of Aquila et al. (2014). They show qualitatively an intensification of the vertical and meridional flows above 10 hPa. Our results show an intensification of the flow below 10 hPa, related to the westerlies, and reduced values of the stream function for the easterlies in the upper stratosphere relative to the control run. The anomalies are smaller in 10Tg30 then in 10Tg60. Overall, this should have consequences for the transport of species like ozone, which are not calculated in this study. However, the streamlines do not represent wave-induced meridional mixing (Butchart, 2014). We show in section 5 that also the quasi-horizontal mixing is important for the transport of sulfate.

## 5   Implication of changes in stratospheric dynamics on the distribution of sulfate

In this section we discuss the relation between the dynamical changes in the QBO and the distribution of sulfate. We discuss how transport and sulfate distribution depend on the QBO phases in section 5 and indicate an impact of the injection rate on transport of sulfate in section 5.

### 5.1   Impact of the equatorial jet structure

Simulation 4Tg60 allows us to examine the differences in transport between different QBO phases. Our definition of QBO phase composites differs from standard definitions in the literature. Typically, the QBO phase is characterized by the equatorial zonal mean wind at a certain level, often 50 hPa, but also levels up to 30 hPa have been used (Baldwin et al., 2001). In this study, QBO phases change due to the impact of sulfate heating, and periods of easterly winds in the lower stratosphere are too rare and weak to base composite on them. Additionally, our aim is to study composites which cover the main characteristics of the equatorial jets under CE and allow to study the impact of QBO phase changes due to CE on transport processes. The chosen composite criterion allows to study the impact of the extended phase of westerly winds in the lower stratosphere on the transport of sulfate and the vertically extended westerly jet in the 30 hPa case and provide the clearest signal of transport differences. We apply the composite criterion for each month of the timeseries and calculate a multi-year monthly mean for each composite:

- Comp West: Westerly winds stronger than 10 m/s at 20 hPa. This composite covers situations in undisturbed QBO and is also close to the situation in 8Tg30.

- Comp East: Westerly winds stronger than 8 m/s at 50 hPa and easterly at 20 hPa. This composite covers many of the westerly tails in 4Tg60 and is close to the situation in 8Tg60.

The criterion for Comp West can be fulfilled under CE (e.g. 8Tg60) but also in an undisturbed QBO. Comp East covers a typical situation under CE conditions but only short periods of an undisturbed QBO. The criteria are chosen to robustly show the impact of CE. Therefore, we also introduce a lower threshold of the zonal wind velocity after testing different composite critria.

In the tropics the meridional distribution of sulfate mass mixing ratios is broader for Comp West with less vertical extension than in Comp East (vertical cross section in Fig. 4, for January (top) and July (bottom)), illustrated also by negative anomalies above 50 hPa in the difference plots (Fig. 4, right column). The differences show higher mass mixing ratios in mid and high latitudes in Comp West which indicates stronger meridional transport in the lower stratosphere. The reason is twofold: wave propagation and different vertical velocity. Following Haynes and Shuckburgh (2000), within westerly QBO winds waves are able to propagate across the equator and break, causing mixing, on the summer side of the westerlies. This results in more meridional mixing into the summer hemi-

January

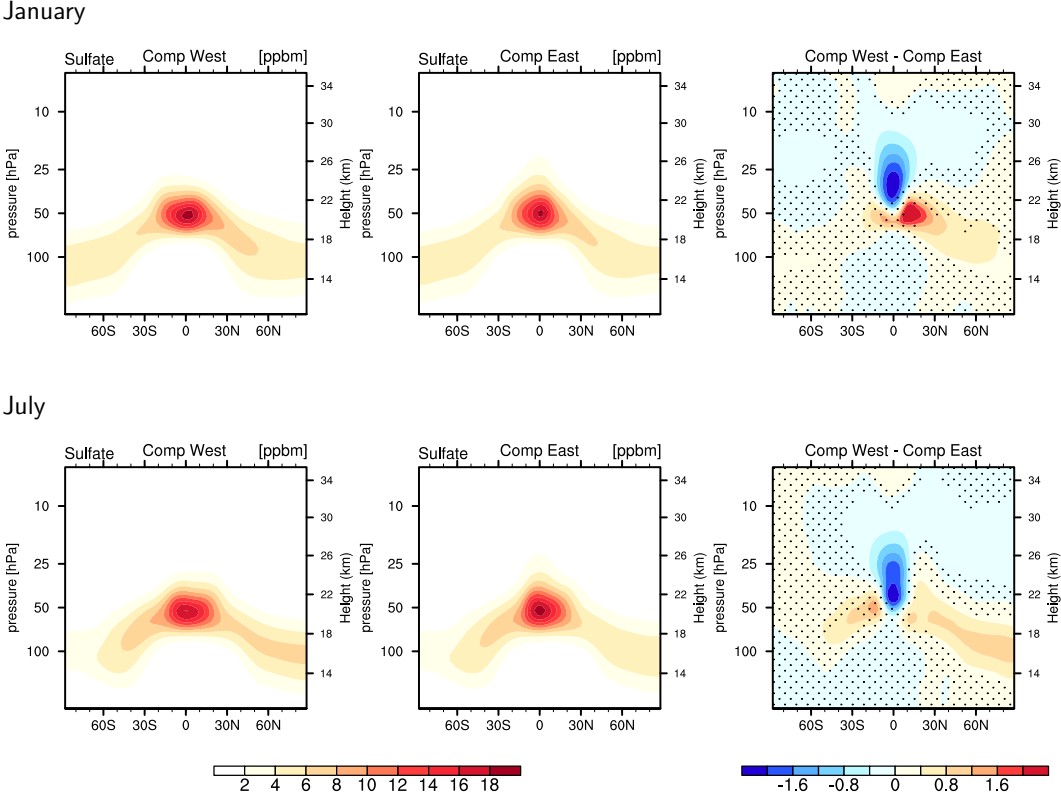

**Figure 4.** Zonal mean of sulfur mass mixing ratio [ppbm] for 4 $\mathrm{Tg(S)yr}^{-1}$ injected at 60 hPa for Composite West (left), Composite East (middle) and difference of Composite W - Composite E (right) for January (top) and July (bottom). Stippling indicates areas which are not significant at the 95% level.

sphere, indicated by higher mass mixing ratios in Comp West in the summerly northern hemisphere in July.

The residual vertical velocity $\omega^*$ (Fig. 5) is similar in both composites below 50 hPa. Above, $\omega^*$ is larger for Comp East, especially around 25 hPa, in both seasons, an area related to the easterly shear zone. This is in agreement with the vertical transport described in Plumb and Bell (1982) for easterly shear. This easterly shear zone overlaps in Comp East with the sulfate layer, which explains the larger vertical extension of sulfate in Comp East. In Comp West the maximum vertical velocity is even stronger but located above 10 hPa, an area with low sulfate mixing ratios.

Figure 6 shows the normalized sulfate burden, i.e. the vertical integral of the $SO_4$ mixing ratio per area, of the two composites as well as the difference of both (right column). The data are normalized by division with the corresponding injection rate. The normalized tropical burden values are slightly lower in Comp West, while the extratropical burden is higher. This is also illustrated by the ratio of tropical to extratropical sulfate burden, which is 1.12 in Comp West, compared to 1.23 in Comp East (Table 1). Additionally, we note an asymmetry in the meridional transport between the hemispheres

January

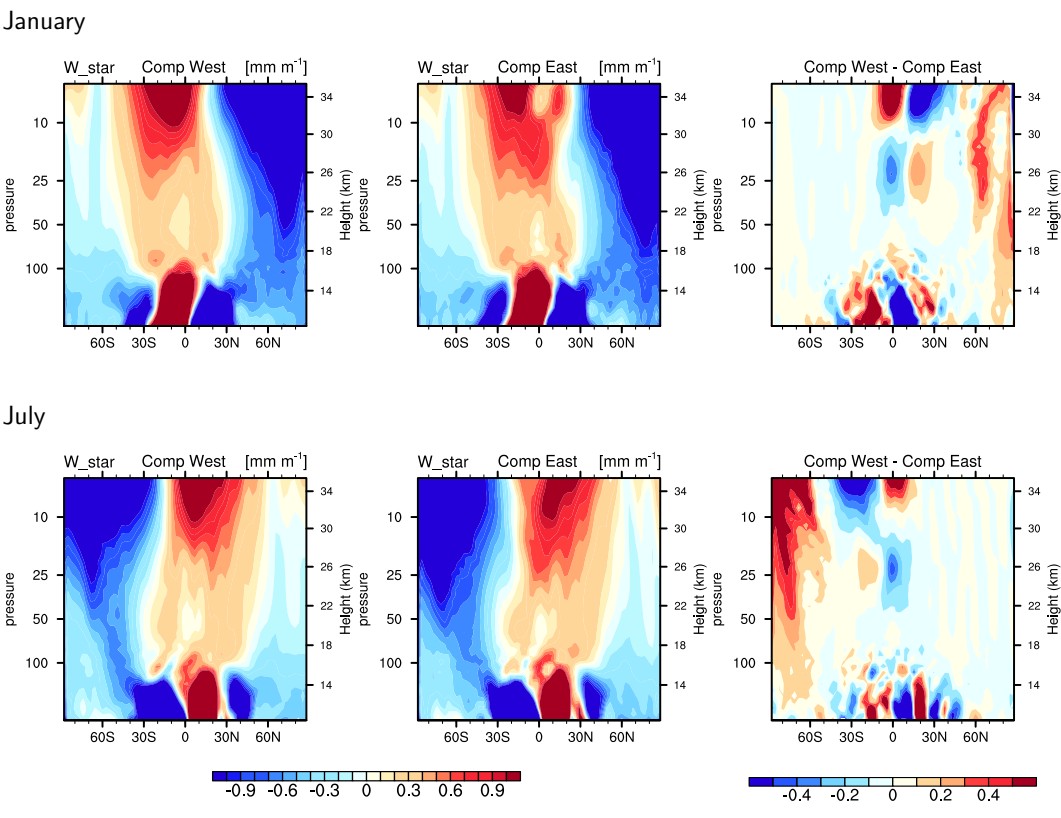

**Figure 5.** Zonally averaged residual vertical velocity $\omega^*$ for both composites (left and middle) of 4Tg60 for January (top) and July (bottom) and the difference of both composites (left).

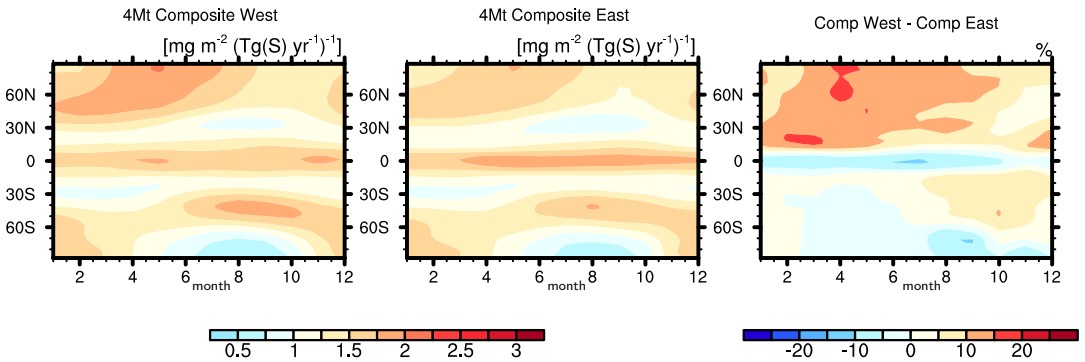

**Figure 6.** Zonal and multi-year monthly means of normalized sulfate burden $[\mathrm{mg\,m^{-2}(Tg(S)\,yr^{-1})^{-1}}]$ of simulation 4Tg60 plotted as Hovmøller diagram. Left: Composite West, Middle: Composite East, and Right: the difference of both in percent.

with up to 20% higher burden in the northern hemisphere in Comp West. This asymmetry is most probably related to the wave activity being stronger in the northern hemisphere, as described above.

 **5.2 Impact of injection rate and height**

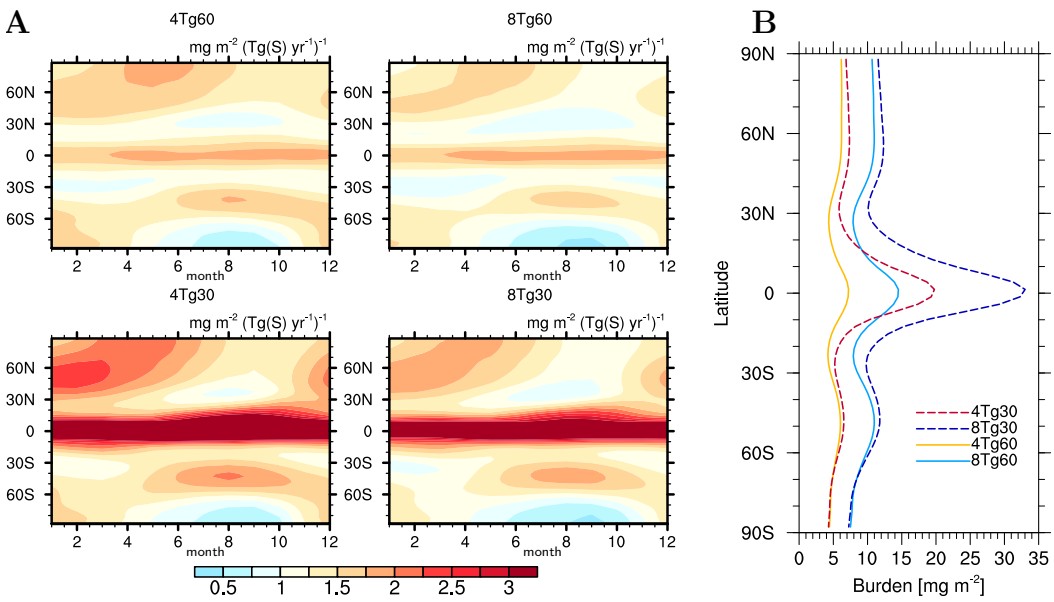

**Figure 7.** A: Zonal and multi-year monthly means of normalized sulfate burden [$\mathrm{mg\,m^{-2}(Tg(S)\,yr^{-1})^{-1}}$], plotted as Hovmøller diagram, for injection rates of $4\,\mathrm{Tg(S)yr^{-1}}$ (left) and $8\,\mathrm{Tg(S)yr^{-1}}$ (right) and injection heights of $60\,\mathrm{hPa}$ (top) and $30\,\mathrm{hPa}$ (bottom). To normalize the data each field is divided by the injection rate. B: Zonally averaged non-normalized sulfate burden.

The Hovmøller diagram of the normalized sulfate burden (Fig. 7, A) shows slightly different patterns of sulfate distribution for the different injection scenarios. All four simulations have common a maximum in the tropics and seasonal variations in the extratropics. Sulfate is accumulated between $40°$ and $60°$ in the winter hemisphere because the winter polar vortex blocks the transport until solar heating breaks down the vortex. In general, in mid-latitudes and high-latitudes the normalized burdens are larger in the $4\,\mathrm{Tg(S)yr^{-1}}$ scenarios than in the 8 Tg scenarios, i.e meridional transport decreases with increasing injection rate. 4Tg60 includes Comp West and Comp East phases. However, by increasing the injection rate we suppress the Comp West phases and the vertical structure of the tropical winds in 8Tg60 is similar to Comp East. Hence, 8Tg60 has less wave induced mixing into the sub-tropics, a stronger tropical pipe, and consequently lower normalized burden in the extratropics than 4Tg60, similar to the differences between the composites in Figure 6. The ratio between the burdens in the tropics and extratropics (Table 1) shows an increase with increasing injection rate from 4Tg60 to 8 Tg60. The decrease in meridional transport intensifies with an increase of the injection rate.

Obvious are the differences between the two injection heights. When injecting at $30\,\mathrm{hPa}$ the trop-
ical maximum increases compared to the $60\,\mathrm{hPa}$ injection results and the area of high sulfate burden
is wider in the tropics: about $20°$ N to $20°$ S instead of $12°$ N to $12°$ S. Also the non-normalized
zonally averaged burden (Fig.7, B) shows for the $30\,\mathrm{hPa}$ cases higher values in the subtropics, re-
lated to the wider jet. Additionally, the subtropical minimum moves poleward (Fig.7, B) while in
the extratropics the non-normalized burden values are similar to the lower injection case. Thus, the
increase in injection height results in stronger burdens mainly in the tropics and subtropics. This dif-
fers from previous results, using a model with lower vertical resolution and no internally generated
QBO, where an increase in injection height results in globally higher burden (Niemeier et al., 2011).
We discuss the impact of these differences on radiative forcing in Section 6.

The higher injection level in simulations 4Tg30 and 8Tg30 extends the sulfate layer vertically.
The westerly jets extend almost up to $5\,\mathrm{hPa}$ making the conditions comparable to Comp West.
Thus, better wave propagation across the equator into the summer hemisphere increases meridional
transport into subtropics (about $20°$ N) compared to 4Tg60 and 8Tg60 in the same way as between
Comp West and Comp East and explains the poleward shift of the subtropical minimum. Punge et al.
(2009) show that the concentration gradient in the subtropics is smaller in the summer hemisphere
during the westerly phase. This causes mixing of sulfate into the sub-tropics which is mixed further
poleward in autumn and results in slightly higher concentrations and burdens in mid-latitudes in
4Tg30 and 8Tg30 compared to the lower injections height. However, this effect is small compared
to the much stronger increase in normalized tropical sulfate burden where both simulations show a
strong maximum. In the $30\,\mathrm{hPa}$ injection case the maximum of the sulfate layer is located at the
level of the stronger transport barrier, causing a stronger confinement in the tropical pipe. Different
to the injection at $60\,\mathrm{hPa}$, the ratio of tropical to extratropical burden is similar for different injection
rates (Tab. 1).

## 6 Implications of changes in stratospheric sulfate transport for radiative forcing

We have shown that radiative heating of the sulfate aerosols impacts the quasi-biennial oscillation
by slowing or even shutting down the oscillation. In turn, the changed QBO impacts the meridional
transport of the sulfate. What does this mean for the efficiency of CE? A good measure for the
efficiency is the TOA radiative forcing. It allows to estimate which forcing can be computed by a
certain sulfur injection. In this study TOA forcing of sulfate is calculated as the difference between
the net TOA flux with aerosols and a TOA flux without aerosols, which is obtained from doubled
radiative transfer calculations (see also Niemeier and Timmreck (2015)).

We perform simulations with an injection rate of $10\,\mathrm{Tg(S)yr^{-1}}$ in order to enable a direct com-
parison to simulation Geo10 of Niemeier and Timmreck (2015). They used ECHAM5-HAM in a
39-layer version that could not simulate an internally generated QBO, but instead constant equa-

torial easterly winds. This allows us to estimate an error in the sulfate forcing made by using a
dynamically too simple model with, additionally, stronger numerical diffusion in vertical direction
due to the larger grid space, which present itself as an additional artificial up- and downdraft. How-
ever, through the distinction of QBO phases in Section 5 we can clearly attribute effects simulated
in this paper to changes in the tropical circulation.

Zonal wind and temperature profiles are similar to the 8Tg results (see also Figure 1 in the sup-
plementary material), while the model resolution without internally generated QBO used in Geo10
simulates easterly winds in the tropics and subtropics. The temperature anomaly is about 1 to 2
K higher in 10Tg60 than in Geo10, but the residual vertical wind velocity is similar at the height
of 50 hPa, the level of the concentration maximum (Fig.;4) and slighly higher in 10Tg30 above
35 hPa, the region of highest mass mixing rations (Figure 2 in the supplementary material).

## 6.1 Effects on aerosol radiative properties

The comparison of simulation Geo10, the low resolution model version without internally gener-
ated QBO, and simulation 10Tg60 shows, in the tropics and sub-tropics, the impact of the different
vertical resolutions. The representation of stratospheric dynamics with constant easterly winds in
Geo10 relates to smaller gradients of potential vorticity (Punge et al., 2009) resulting in a stronger
meridional transport. This indicates also the lower ratio of tropical to extratropical mean burden is
lower in Geo10 than in 10Tg60 (Table 1). The model version with lower vertical resolution tends
to overestimate meridional transport, a tendency also seen in earlier volcano studies (Niemeier et al.
(2009), Timmreck et al. (1999)).

A clear effect is the roughly 20% lower AOD in 10Tg60 compared to Geo10 outside of the tropics
with similar burden values in this region. The AOD is a measure of turbidity and degradation of
sunlight and depends, for sulfate, on the particle size. Small particles scatter more efficiently than
larger particles. The effective radius increases from 0.4 µm in Geo10 to 0.45 µm in 10Tg60 (Fig. 9)
and, consequently, the AOD decreases. The better representation of the stratospheric dynamics in the
higher resolution simulations and the resulting stronger confinement of the particles in the tropics
cause them to grow larger.

Increasing the injection height to 30 hPa, simulation 10Tg30, further intensifies the equatorial
confinement which is stronger at a height of 30 hPa than at a height of 60 hPa. Thus, injection
at 30 hPa results in strong tropical maxima of burden and AOD, as discussed in Section 5. Small
particles with little sedimentation are transported vertically in the tropical pipe and meridionally in
the upper branch of the BDC while coarse mode particles are transported in the lower branch (Fig. 2
supplementary material). In the extratropics the AOD is 30% to 50% lower compared to simulation
10Tg60. Due to the stronger tropical confinement the particles grow to radii up to 0.75 µm, an
increase of about 0.25 µm compared to 10Tg60. The reduction of AOD in the extratropics is strong
enough to reduce the global mean AOD of 10Tg30 compared to 10Tg60 (Table 1).

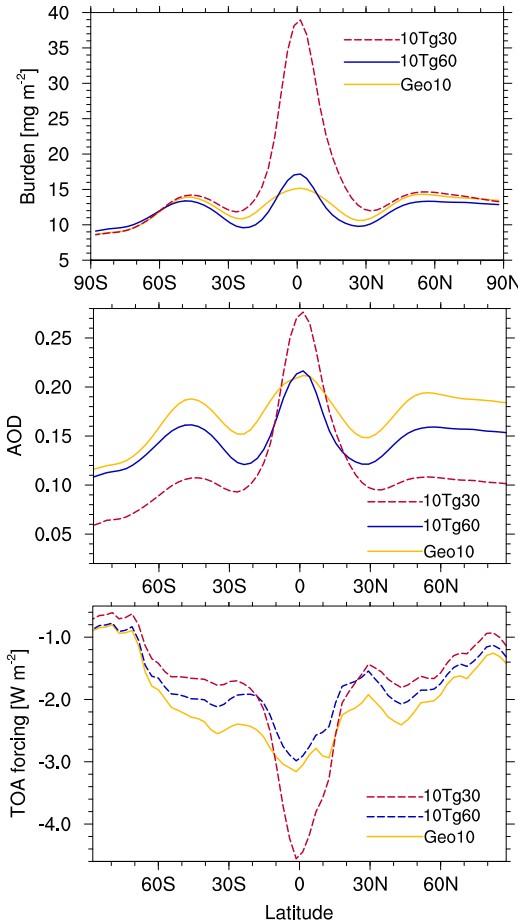

**Figure 8.** Zonal mean sulfate burden (top), aerosol optical depth at 550 μm (middle) and top of the atmosphere forcing (bottom) for different experiments with injection rates of $10\,\mathrm{Tg(S)yr}^{-1}$.

## 6.2 Effects on radiative forcing

Niemeier and Timmreck (2015) have shown that the global TOA radiative forcing depends on the meridional distribution of aerosols. Models which simulate a stronger tropical confinement (e.g. English et al. (2012)) show lower TOA forcing per injected amount of sulfur. So the increased tropical
confinement and decreased AOD in the extratropics in simulations with internally generated QBO and the described shift of particle size towards larger particles should change the forcing efficiency, calculated as the ratio of TOA radiative forcing and injection rate, as discussed in Niemeier and Timmreck (2015).

    The zonal mean of TOA radiative forcing is smaller in simulation 10Tg60 than in Geo10 (Figure
8, bottom), even close to the equator, where the AOD of 10Tg60 is slightly larger. This is again the impact of the shift to larger particles.

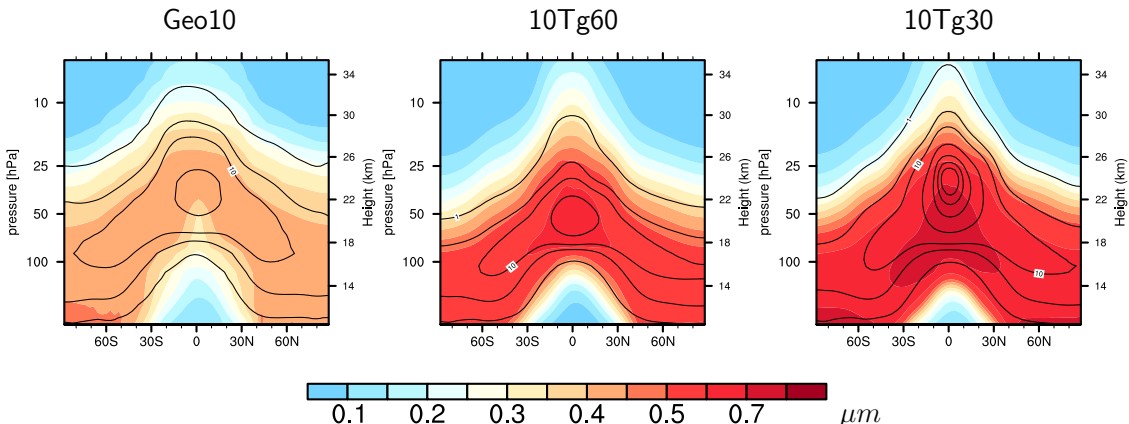

**Figure 9.** Zonal mean of effective radius [μm] of sulfate aerosols for experiments Geo10 (left), 10Tg60 (middle), and 10Tg30 (right) with an injection rate of $10\,\text{Tg(S)yr}^{-1}$. Contour lines for sulfate aerosol mass mixing ratio [ppm] are plotted for 1, 5, 10, 25, 50, 75, 100 ppm.

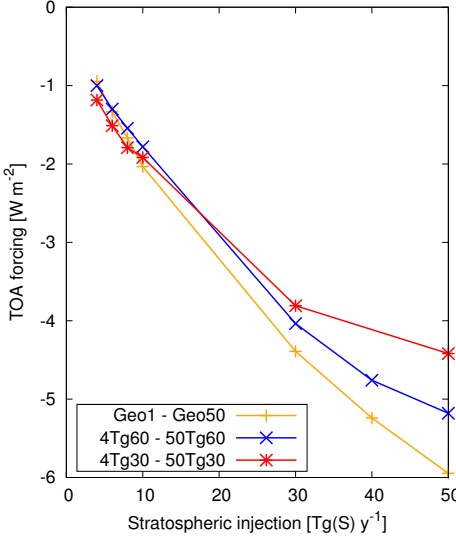

**Figure 10.** Injection rate against globally averaged top of the atmosphere radiative forcing (all sky) for simulations with low vertical resolution (GeoX) and high vertical resolution and two injection heights (XTg60 and XTg30).

Comparison of the global TOA forcing (Figure 10 and Table 2) of sulfate of this study (XTg60, blue line) to simulations with lower vertical resolution (GeoX, orange line) of Niemeier and Timmreck (2015) indicate a smaller increase with increasing injection rate in the Xtg60 simulations. Thus, 420 the efficiency of the sulfur injection with increasing injection rate decreases stronger than described in Niemeier and Timmreck (2015). While forcings are very similar for lower emission rates, for in-

jection rates above $10\,\mathrm{Tg(S)yr^{-1}}$ the forcing in the XTg60 simulations is 10% to 13% lower. This would require even stronger injection amounts to counteract a certain greenhouse gas forcing.

Previous studies showed an increasing efficiency with increasing injection height. In Niemeier and Timmreck (2015) TOA forcing increases by 50% for an injection of $10\,\mathrm{Tg(S)yr^{-1}}$ when changing the injection height from $60\,\mathrm{hPa}$ to $30\,\mathrm{hPa}$. In this study, TOA forcing of 4Tg30 increases only by 18% and by 8% for 10Tg30 compared to 4Tg60 and 10Tg60, respectively. The efficiency even decreases for strong injection rates of $20\,\mathrm{Tg(S)yr^{-1}}$ and more as a consequence of the strong tropical confinement in the high injection cases. This result challenges an injection at $30\,\mathrm{hPa}$ at the equator, because, additionally, it is technically much more demanding (Moriyama et al., 2016).

**Table 2.** Top of the atmosphere radiative forcing $[\mathrm{Wm^{-2}}]$, calculated using double radiative transfer calculations, for simulations with a 39-layer version of the model and injection height at $60\,\mathrm{hPa}$ (GeoX) and two 90-layer model simulations with injection height at $60\,\mathrm{hPa}$ (XTg60) and $30\,\mathrm{hPa}$ (XTg30).

| Simulation | Injection rate $[\mathrm{Tg(S)yr^{-1}}]$ | | | | | | |
| --- | --- | --- | --- | --- | --- | --- | --- |
| | 4 | 6 | 8 | 10 | 30 | 40 | 50 |
| GeoX | -0.95 | -1.33 | -1.67 | -2.03 | -4.39 | -5.24 | -5.947 |
| XTg60 | -1.0 | -1.29 | -1.54 | -1.78 | -4.04 | -4.76 | -5.18 |
| XTg30 | -1.18 | -1.51 | -1.79 | -1.92 | -3.81 | | -4.42 |

## 6.3 Effects of wider injection area

To test how specific choices of the injection area may alter the effect of the QBO, the injection area is increased to a band between $30°\,\mathrm{N}$ and $30°\,\mathrm{S}$ (10Tg60lat30). Injecting partly into the surf-zone increases meridional transport and reduces the amount of sulfur injected in the tropics. This reduces the impact on the QBO and causes no longer a complete shut down of the QBO but an extended period of the oscillation of roughly five years (Fig. 11).

This simulation results in lower AOD in the tropics, but up to 50% higher values in the extratropics compared to 10Tg60 (Fig. 12, left). The maximum of the AOD is shifted into the extratropics because less sulfate is confined in the tropics. The resulting radiative forcing has its maximum around $40°\,\mathrm{S}$ (Figure 12, right), caused by stronger transport into the southern hemisphere. In 10Tg60lat30 the model simulates long periods with easterly shear (Figure 11), similar to the conditions of Comp East, which also results in stronger transport into the southern hemisphere (Figure 6).

Extending the injection area reduces the impact on the QBO, and leads to decreased forcing in the tropics and increased forcing in mid-latitudes. A further increase of the injection area would likely strengthen this effect. Another option would be to inject poleward of $\pm 15°$. This would leave the tropics with much lower reduction in solar radiation because almost no extratropical air is directly mixed into the tropics in the stratosphere (O'Sullivan and Chen, 1996), only by wave induced mixing as described in Section 5. Laakso et al. (2017) found 40% lower radiative forcing in the tropics

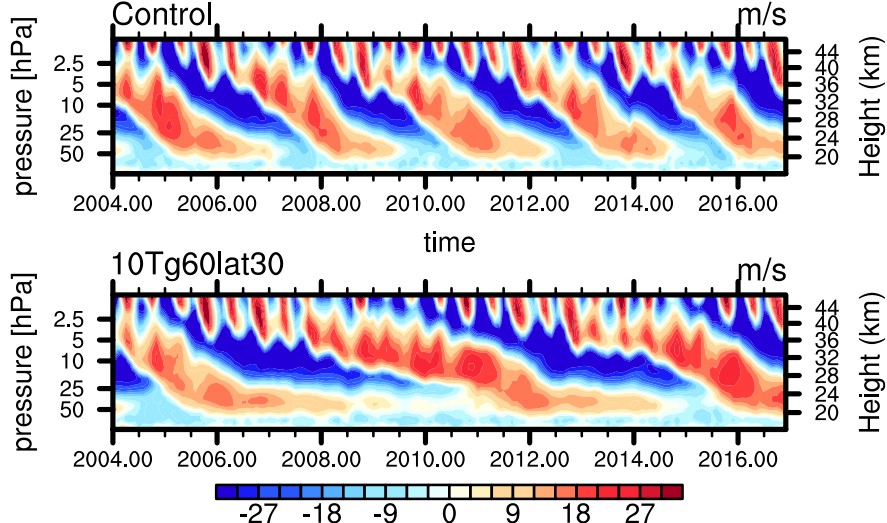

**Figure 11.** Zonal mean zonal wind velocity at the equator for the control simulation and for an injection of $10\,\mathrm{Tg(S)yr^{-1}}$r between $30°$ N and $30°$ S.

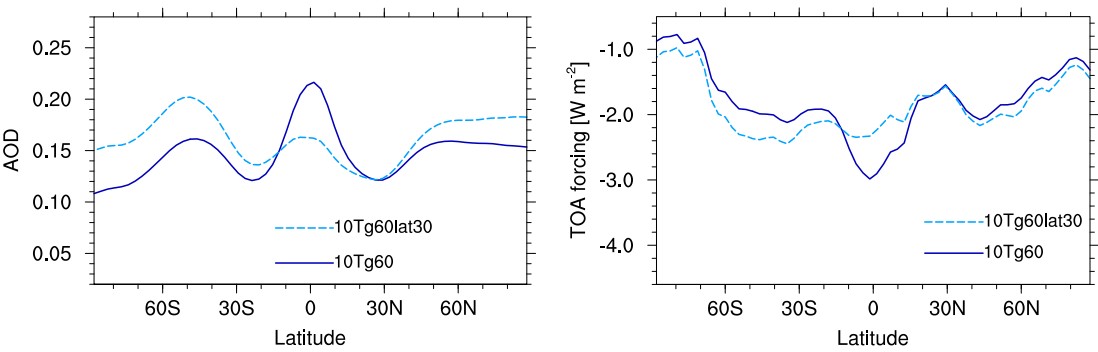

**Figure 12.** Aerosol optical depth at 550 μm (left) and top of the atmosphere forcing (right) for an experiment with injections between $30°$ N and $30°$ S (10Tg60lat30) and 10Tg60.

when injecting at $15°$ from the equator compared to an equatorial injection. We show that the meridional transport depends on the impact of the sulfate heating on the equatorial winds in the tropical stratosphere. Thus, the injection strategy may play an important role in the global distribution of the sulfate aerosol. The climatic impact of the aerosol distribution in 10Tg60lat would differ from previous studies with a globally more homogeneous CE forcing distribution.


## 7 Conclusions

The results of this study show a strong impact of the absorption of infrared radiation by sulfate and the related additional warming in the stratosphere on the dynamics and transport processes in this region. Our results differ in detail from Aquila et al. (2014) but confirm their results on the impact of the stratospheric heating on the QBO. The dynamical state of the stratosphere determines the transport of species from the tropics into the extratropics. Prolonged phases of westerly winds in

the lower stratosphere develop for an injection rate of $4\,\mathrm{Tg(S)yr}^{-1}$, and a shut down of the QBO occurs for injections of $6\,\mathrm{Tg(S)yr}^{-1}$ and more. In our simulations with an injection height of $60\,\mathrm{hPa}$ tropical confinement of the sulfate gets stronger with increasing injection rate. In the results of an injetion height at $30\,\mathrm{hPa}$ this feature is most propably masked by larger particle size and, thus, stonger sedimentation. The consequence is a decreased meridional transport out of the tropics and

decreased AOD and TOA forcing in the extratropics. Moderate westerly winds in the tropics at the height of the sulfate layer cause stronger transport towards the northern hemisphere (Section 5). Easterlies or westerly vertical shear, as e.g. dominating in the wider injection case (10Tg60lat30), cause stronger transport towards the southern hemisphere (Section 6.3). We assume the reason is stronger wave propagation through the tropics into the subtropics in westerly winds (Section 5).

This different stratospheric transport would impact also dynamical processes and the hydrological cycle in the troposphere. Haywood et al. (2013) describe, e.g., a strong shift of the position of the ITCZ when shielding only one hemisphere via a sulfate layer, which is likely after an extratropical volcanic eruption.

A vertically extended sulfate layer results from injections at $30\,\mathrm{hPa}$. The consequence is a strong

westerly jet, which extends high into the stratosphere ($5\,\mathrm{hPa}$). Meridional transport in our strong and high level injection case (10Tg30) is reduced to a point that CE would impact much stronger the tropics than subtropics. Previous simulations indicated a strong increase of AOD and TOA radiative forcing (up to 50%) when increasing the injection height (English et al. (2013), Niemeier and Timmreck (2015)). In this study, we obtain only a small increase of 18% for $4\,\mathrm{Tg(S)yr}^{-1}$, of 8%

for $10\,\mathrm{Tg(S)yr}^{-1}$ and even less forcing than the lower injection height for strong injections. Both previous studies were performed with models not generating a QBO. This shows the importance of a realistic representation of stratospheric dynamics, in particular of the QBO, for the QBO relevant transport patterns in transport studies like CE, evolution of volcanic sulfate, and studies of stratospheric chemistry. A conclusion from this result can be that injecting at high levels at the equator

might be unfavorable for CE, not only because it is technically more demanding.

Our study shows that transport of aerosol, and other species, in the tropical and sub-tropical stratosphere is complex as it depends not only on the season but also on the QBO phase and, thus, on the structure of the equatorial jets. Lagrangian tracer studies of Punge et al. (2009) show quite different transport of a tracers depending on the emission region and QBO phase. Further, the emission strat-

egy may vary, which changes the sulfate heating impact on the QBO and consequently the transport.

Interaction of radiatively active species, like ozone, with sulfate may also impact the structure of the equatorial jets (Richter et al., 2017).

The simulations in this study do not include stratospheric chemistry. Therefore, we can not describe the impact of the changes in the stratosphere on other chemical species like ozone or methane. Ozone would be impacted twofold: via chemical reactions related to sulfur chemistry and via changed transport. The described reduction in meridional transport of sulfate may also be true for ozone and the stratospheric ozone concentration could be reduced in the extratropics. This reduction would add to the proposed reduction of ozone due to chemical reactions (Tilmes et al., 2008). Pitari et al. (2016) discuss the impact of volcanic aerosol on stratospheric dynamical processes and calculate a reduction of approximately 10% of the extratropical mass flux of $NO_2$ and $CH_4$ after the eruption of Mt. Pinatubo. Changes in ozone and other radiatively active gases may of course also feed back on the dynamics. Such effects are not covered by our study.

An additional caveat is the fact that the sea surface temperatures in our simulations are prescribed independently of the emission scenario. Estimated from GeoMIP simulations (Niemeier et al., 2013), CE with an injection of $6 \, \mathrm{Tg(S)yr^{-1}}$ would roughly cause a decrease of the sea surface temperature of $1\,^\circ\mathrm{K}$. This impacts convection which then modifies the generation of gravity waves and likely the period of he QBO. Such effects should be assessed in future studies.

In this study we calculated a smaller efficiency of sulfur injections than Niemeier and Timmreck (2015) obtained in model simulations with lower vertical resolution and, hence, less realistic tropical dynamics. Therefore we have to modify some of the conclusions drawn in Niemeier and Timmreck (2015). They estimated an injection of $45 \, \mathrm{Tg(S)yr^{-1}}$ would counteract global greenhouse gas forcing of $6 \, \mathrm{W/m^{-2}}$. This amount would be necessary to keep the global mean temperature at 2020 level in 2100 while maintaining business as usual emissions. The decreased forcing efficiency simulated in this study would increase the injected amount to $70 \, \mathrm{Tg(S)yr^{-1}}$ to be injected for such a forcing. Adapting a strategy of Laakso et al. (2017) with injections following the zenith of the sun or injecting at 15°N and 15°S, may slightly reduce the injection rate. However, the spread in the forcing simulated by different models is large (Niemeier and Tilmes, 2017), as is the amount of injected sulfur necessary to generate a certain forcing. Estimates of lifting costs of sulfur into the stratosphere (e.g. (Moriyama et al., 2016)) depend strongly on the efficiency of the injection.

Finally, it needs to be stated that the simulated impact of stratospheric sulfate heating on the QBO is only a model result which cannot be evaluated in reality. However our simulations further show that the efficiency of sulfur injections may depend crucially on the jet structure in the tropical stratosphere, which itself will be influenced strongly by the injections. Our simulations show that the dynamical effects vary strongly even in different configurations of the same model. To reduce this uncertainty a better understanding of tropical dynamics and model simulations without the necessity of gravity wave parameterizations, i.e. with horizontal resolutions at least one order of magnitude higher than used here, may be necessary. As for many questions related to CE, certainty of response

would require the full implementation of CE. It would be nice to confirm the effect of sulfate aerosols on the QBO in observations after volcanic eruptions, but this is difficult due to the small number of well observed large tropical eruptions, the short lifetime of volcanic aerosols, and the internal variability of the QBO.

*Acknowledgements.* We thank two anonymous reviewers for their helpful comments, Simone Tilmes and Yaga Richter for inspiring discussions, Marco Giorgetta for valuable comments at different stages of the paper and Rene Hommel, Jan Kazil, Harri Kokkola and Hanna Vehkamäki for their earlier help to modify HAM. This work is a contribution to the German DFG-funded Priority Program 'Climate Engineering: Risks, Challenges, Opportunities?' (SPP 1689). U. Niemeier is supported by the SPP 1689 within the project CEIBRAL and CELARIT. Earlier parts of the study were funded by the German Environment Agency (UBA). The simulations were performed on the computer of the Deutsches Klima Rechenzentrum (DKRZ). Information on model data used in this publication is available at http://www.mpimet.mpg.de/en/staff/ulrike-niemeier/geoengineering/data/

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
