# Peer review of "Changing transport processes in the stratosphere by radiative heating of sulfate aerosols"

_Atmospheric Chemistry and Physics, 2017_

## Referee Comment (RC1) · Anonymous Referee #1 · 31 May 2017

This manuscript investigates the effect of stratospheric sulphate aerosols on stratospheric transport, and, importantly, the effect of correct simulation of these transport processes to the climate effects of geoengineering using these aerosols. The manuscript provides further explanation as to why the radiative forcing of sulphate aerosols does not increase linearly with injection rate, and explains why the efficiency of the forcing per unit injection rate may decrease even more sharply than previously estimates. The manuscript is well-organised and mostly clearly written. Some of the figures could be improved and there are some issues with the clarity of the text to correct (see below). On the whole these constitute minor revisions to the manuscript, after which I recommend the manuscript be accepted.

Main points:

1. One area I would like to see a little more discussion of is the choice of the criteria for the QBO composites. They seem somewhat arbitrary. I would like to see some more justification of the choices the author made and some discussion of the importance of these choices. Key questions for me include: How were they arrived at? Were a range of other values for the criteria tested? Are the results sensitive to these choices?

2. It would be useful to have a table summarising the forcing efficiencies of the different simulations. This is all discussed in the text but it would be helpful to the reader to have some of the key statistics drawn out in the form of a table, especially since the authors rightly highlight the efficiencies as a key implication of the study.

Clarity of language:

L73 - are the terms 'tropical' and 'equatorial' jets being used interchangeably here?

L84 - related **to**

L131 - I think 'imagine' would be a better choice of word than 'assume' here

L176 - month –> months

L185 - 'causes a prolongation' –> 'prolongs'

L222 - 'the vertical extension of the jet' - which jet is being referred to here? The equatorial jet, I presume?

L430 - 'thought' –> 'through'

L455 - missing the units of temperature

Figures:

1. Figure 1 is missing units on the axes.

2. Figure 2 is missing a title and units on the 'pressure' axis. The font size could also be increased to make it easier to read.

---

## Referee Comment (RC2) · Anonymous Referee #2 · 2 Jul 2017

Review of "Changing transport processes in the stratosphere by radiative heating of sulfate aerosols" by U. Niemeier and H. Schmidt

This paper focuses on addressing changes in sulfate aerosol transport in the stratosphere in the presence of different states of the quasi-biennial oscillation (QBO). The paper addresses a very important and timely topic, however the conclusions presented are not supported by the figures shown, and there is ambiguity in some of the presented analysis. Therefore, I recommend major revisions before this manuscript can be published.

Major Comments:

[Figure]

1. The methodology of compositing QBO phases, beginning of Section 5.1 is not clear. Transport varies with QBO phase as well as with the month of the year. The discussion in lines 260 – 270 implies that the QBO phase distinction is made for each month (correct), however it seems that in Figure 4, all months with QBOW and all months with QBOE are composited together. The caption does not say anything about which months are used in the composite, so I'm assuming that all months are used. If this is indeed the case, the plot does not show differences between QBOE and QBOW, but it shows those differences as well as differences between compositing different months of the year that coincide with QBOE and QBOE, and therefore does not answer the question that was posed. If the authors are indeed looking at the month of January or July in this Figure (the only way in which the differences between QBOE and QBOW phases can be separated clearly), then this needs to be clarified.

2. The authors in section 6 compare a simulation with 10T(s)/yr carried out with a 90-level (with QBO) and 39-level (without QBO) versions of the model in order to further demonstrate effects of the QBO. However, at no point in the manuscript do the authors show or discuss that properties of the models critical to aerosol transport are the same between the 39 and 90-level versions (except the QBO). For example, is the mean Brewer-Dobson circulation the same between these models? Is the dissipation from planetary waves (that directly impacts mixing) the same? What is the strength of stratospheric DJF and JJA jets? Are the temperature anomalies due to injections the same in both models? Without these aspects of the model demonstrated to be the same (or very similar), it is not clear here whether we're looking at differences due to the QBO or differences due to other model differences. I suggest that similarities between the aspects of the models mentioned above are shown in the appendix.

3. Interpretation of differences between simulation 10Tg60 and Geo10 I and 10Tg30 (Figures 7 and 8) is not clear and very confusing. There is no convincing explanation in the text to account for the differences in AOD between the 3 simulations shown in Figure 7. There is an overall increase in aerosol number density in the accumulation mode

in Geo10 as compared to 10Tg60 (Figure 8), however if the contour level is adjusted to show the maximum contour, it could be that the distribution with latitude of aerosols looks very similar to that as in 10Tg60, and the overall change is due to something other than QBO winds. Secondly, if the westerly QBO is inhibiting transport out of the tropics, why doesn't this apply to 10Tg30? There are plenty of aerosols in the accumulation mode in the extratropics and the QBO is in an even stronger westerly phase. Besides, if QBOW intensified vertical transport as mentioned earlier in the manuscript, why is the distribution of coarse mode particles in Figure 8 for 10Tg60 (bottom center panel) confined to a smaller vertical region as compared to that in Geo10 (left bottom panel). This plot alone implies higher vertical velocity at the equator in Geo10 as compared to 10Tg60. Again, this difference could be the consequence of different model dynamics, wave breaking and BD circulatoin due to differences in vertical resolution, and not to the QBO. Authors state is line 360 that the number density in the coarse mode is lower in Geo10 then in 10Tg60. I don't see that from Figure 8 – that maybe true right at the equator in a very small region, but overall the number density in the coarse mode if anything is bigger in Geo10.

4. Interpretation of differences in the text between 10Tg60 and 10Tg30 are also not consistent with the figures. For example: Lines 350-351: The difference between 10Tg60 and 10Tg30 is primarily the injection altitude and not the tropical wind system From Figure 10Tg60 shows that likely the aerosol transport out of the tropics occurs via the lower branch of the BDC, where in 10Tg30, the aerosols are transported via mixing and the upper branch of the BDC (see also minor comment 1).

Lines 366- 367: 'Figure 8 indicates low meridional transport resulting in low particle number densities in extratropics in 10Tg30.' This is not at all consistent with top rightmost panel in Figure 8: there are plenty of particles in the extratropics.

5. In order to explain the differences between AOD in the 3 simulations shown in Figure 7, it would be helpful to plot effective radius of particles and surface area density, instead of what is currently in Figure 8.

Minor Comments:

1. Line: 77-78: "This quasi-horizontal mixing is the main transport branch for sulfate aerosols'. This statement highly depends on the location of injection of the aerosols. There are three main ways the aerosols can be transported out of the tropics: a) The deep branch of the BDC, the shallow branch, and horizontal mixing. Aerosols injected right above the tropical tropopause are mostly going to be transported with the shallow branch of the BDC, those injected several kilometers above the tropopause will likely be primarily transported with the upper branch of the BDC. Some will be transported horizontally by mixing. I suggest that a discussion of the different branches of the BDC is added and how the location of injection (30 hPa and 60 hPa discussed here) affect which branch of the BDC is the primary transport mechanism. Figure 1 of Bonisch et al. 2011 has an excellent graphic (Atmos. Chem. Phys., 11, 3937–3948, 2011 www.atmos-chem-phys.net/11/3937/2011/ doi:10.5194/acp-11-3937-2011)

2. Line 92: An average period of the QBO is 28 (not 29) months.

3. Figure 1: Why isn't the QBO included here for the 4 Tg 30 hPa injection? Please include it.

4.Temperature anomalies in rightmost panel of Figure 2 clearly exceed the colorbar. Please change the colorbar so the maximum and minimum temperature anomalies are clear in all the panels. Please in the text also include the amplitude of maximum temperature anomalies for the simulations in Figure 2.

5. Line 203-204: 'Positive anomaly does not extend to the pole . . . because polar vortex blocks the transport' - it would be helpful to overplot the aerosol concentrations here to demonstrate this point clearly.

6. Why is there such a strong negative temperature anomaly near 5 hPa at the equator in the 8Tg30 hPa simulation? (Rightmost panel of Figure 2)

7. Figure 3: The color-scale is inappropriate. It is impossible to see what are the zonal

wind velocities in the top panels as well as anomalies in the bottom panels. Both clearly exceed the color scale. Please correct.

8. Figure 8: Again here, the colorbar needs to be adjusted that it is clear what the maximum contour is in the top leftmost panel.

9. Line 445-447: That is too strong of a conclusion! Injecting at 30 hPa and other location could be viable at other latitudes – only equatorial injections have been shown here, hence authors should not make such a sweeping conclusion.

10. Lines 457- 469: I'm not sure how this paragraph is relevant to the main point of the study. If the authors chose to keep it, please explain how you arrived at the injection estimates up to 2100 mentioned in line 460 and 462.

---

## Author Comment (AC1) · 12 Sep 2017

**Answers to reviewer 1 on the ACPD paper (acp-2017-470)**
Changing transport processes in the stratosphere by radiative heating of sulfate aerosols

Ulrike Niemeier and Hauke Schmidt
Max Planck Institute for Meteorology, Bundesstr. 53, 20146 Hamburg, Germany

We thank the reviewer for the helpfully comments. We added some word of explanation to the description of the composites, changed the font sizes in many figures and added the table of top of the atmosphere forcing values.
Cited text is given in *italic* and new text in blue.

**1. One area I would like to see a little more discussion of is the choice of the criteria for the QBO composites. They seem somewhat arbitrary. I would like to see some more justification of the choices the author made and some discussion of the importance of these choices. Key questions for me include: How were they arrived at? Were a range of other values for the criteria tested? Are the results sensitive to these choices?**

We are interested in the impact of the changes in the QBO due to climate engineering on the transport of sulfate. Thus, the question behind the composite criteria of the 4Tg60 simulation was to find a criterion which indicates the impact of the prolonged phases of westerly winds in the lower stratosphere. This phase was important in 4Tg60 and even more in 8Tg60.
We tested some variations of the final composite. The final composites gave the clearest signal on differences in transport. We introduced also lower thresholds for the wind velocity of a composite to get clear signals. Therefore the results are robust to small variations in the composite. However, a criterion for easterly winds in the lower stratosphere, as e.g. described in Baldwin (2001) should give a different result. But periods with easterlies at 50 hPa were short and wind velocity weak. A Hovmoeller diagram for this criterion would have been statistically not well based. As the simulations were time consuming, we were not be able to continue the simulation for more than 50 years.
We chang the text to:
*Simulation 4Tg60 shows still changing QBO phases, e.g. periods with easterly winds in the equatorial lower stratosphere or phases of easterly shear. This allows to examine the differences in transport between different QBO phases. Our definition of QBO phase composites differs from usual definitions in the literature.* Typically the QBO phase is defined by using the equatorial zonal mean wind at a certain level, mostly 50 hPa, but also levels between 45 hPa and 30 hPa are common (Baldwin, 2001). In this study QBO phases change due to the impact of sulfate heating and periods of easterly winds in the lower stratosphere are too rare and weak to base composite onto them. Additionally, our aim was to get composites which cover the main characteristics of the equatorial jets under CE and allow to study the impact of QBO phase changes due to CE on transport processes. The chosen composite criterion allows to study the impact of the extended phase of westerly winds in the lower stratosphere on the transport of sulfate and the vertically extended westerly jet in the 30 hPa case and gave the clearest signal on differences in transport. *We apply the composite criterion for each month of the timeseries and calculated a multi-year monthly mean for each composite:*

- *Comp West: Westerly winds stronger than 10 m/s at 20 hPa. This composite covers situations in undisturbed QBO and is also close to the situation in 8Tg30.*

- *Comp East: Westerly winds stronger than 8 m/s at 50 hPa and easterly at 20 hPa.* This composite covers many of the westerly tails in 4Tg60, easterly shear, and the jets in 8Tg60.

*The criterion for Comp West can be fulfilled under CE (e.g. 8Tg60) but also in an undisturbed QBO. Comp East covers a typical situation under CE conditions but only short periods of an undisturbed QBO. The criteria are chosen to robustly show the impact of CE. Therefore, we also introduce a lower threshold of the zonal wind velocity after testing different composite critria.*

**2. It would be useful to have a table summarizing the forcing efficiency of the different simulations. This is all discussed in the text but it would be helpful to the reader to have some of the key statistics drawn out in the form of a table, especially since the authors rightly highlight the inefficiencies as a key implication of the study.**
We add the table in Section 6.2:

Table 1: Top of the atmosphere radiative forcing [W m$^{-2}$], calculated from a radiation double call, for simulations with a 39-layer version of the model and injection height at 60 hPa (GeoX) and two 90-layer model simulations with injection heights at 60 hPa (XTg60) and 30 hPa (XTg30).

| Simulation | Injection rate Tg(S) yr$^{-1}$ | | | | | | |
|---|---|---|---|---|---|---|---|
| | 4 | 6 | 8 | 10 | 30 | 40 | 50 |
| GeoX | -0.95 | -1.33 | -1.67 | -2.03 | -4.39 | -5.24 | -5.95 |
| XTg60 | -1.00 | -1.29 | -1.54 | -1.78 | -4.04 | -4.76 | -5.18 |
| XTg30 | -1.18 | -1.51 | -1.79 | -1.92 | -3.81 | | -4.42 |

**Minor comments**

**L73 - are the terms 'tropical' and 'equatorial' jets being used interchangeably here?**
Yes. We changed tropical jet to equatorial jet to avoid confusion.

**L84 - related 'to'**
**L131 - I think 'imagine' would be a better choice of word than 'assume' here**
**L176 - month –> months**
**L185 - 'causes a prolongation' –> 'prolongs'**
All done.

**L222 - 'the vertical extension of the jet' - which jet is being referred to here? The equatorial jet, I presume?**
We added equatorial jet in the sentences as well as in the previous sentence.
**L430 - 'thought' –> 'through' L455 - missing the units of temperature**
All done.

**Figures**

**1. Figure 1 is missing units on the axes. 2. Figure 2 is missing a title and units on the 'pressure' axis. The font size could also be increased to make it easier to read.**
Done, also for the following figures.

---

## Author Comment (AC2) · 12 Sep 2017

**Answers to Reviewer 2 on the ACPD paper (acp-2017-470)**
Changing transport processes in the stratosphere by radiative heating of sulfate aerosols

Ulrike Niemeier and Hauke Schmidt
Max Planck Institute for Meteorology, Bundesstr. 53, 20146 Hamburg, Germany

We thank the reviewer for the helpful comments. We try to clarify the interpretation of the results. We include now a figure of the effective radius and move the particle number density to supplementary material. Within the supplementary material we additionally show zonal wind, residual vertical velocity, and temperature of the 10 Tg(S)y$^{-1}$ injection cases as asked by the reviewer. Accordingly, we change the discussion in the text. To strengthen our argumentation on the transport into the extratropics we add the ratio of sulfate burden in the tropics and extratropics. The ratio increases with injection rate mainly in the 60 hPa injection case, less in the 30 hPa injection case, indicating increased confinement in the 60 hPa injection.

We also add citations to previous studies which compare the impact of the vertical resolution on BDC, age of air etc. for different vertical resolutions of the mode. We say in the text that the differences between the different model resolutions can be related to the QBO but also to different vertical numerical diffusion.

Citations of the text are written in *italic* and changed or new text is highlighted in blue.

**1. The methodology of compositing QBO phases, beginning of Section 5.1 is not clear. Transport varies with QBO phase as well as with the month of the year. The discussion in lines 260 – 270 implies that the QBO phase distinction is made for each month (correct), however it seems that in Figure 4, all months with QBOW and all months with QBOE are composited together. The caption does not say anything about which months are used in the composite, so I'm assuming that all months are used. If this is indeed the case, the plot does not show differences between QBOE and QBOW, but it shows those differences as well as differences between compositing different months of the year that coincide with QBOE and QBOE, and therefore does not answer the question that was posed. If the authors are indeed looking at the month of January or July in this Figure (the only way in which the differences between QBOE and QBOW phases can be separated clearly), then this needs to be clarified.**

We add additional explanation to the definition of the composites (see also Answers to reviewer 1). We calculate the composites separately for each month but plot mostly annual mean values for zonal averages and Hovmoeller diagrams for the sulfate burden. We follow the reviewers advise and plot the months of January and July instead of the annual mean in Figure 4. Additionally, we split the plot into one for the sulfate concentration and one for the vertical velocity. Overall, the result stays the same.

The description of the different transport in both composites change to: *In the tropics the meridional distribution of sulfate mass mixing ratio is broader for Comp West with less vertical extension than in Comp East (vertical cross section in Fig. 4, for January (top) and July (bottom)), illustrated also by negative anomalies above 50 hPa in the difference plots (Fig. 4, right column). The differences show higher mass mixing ratios in mid and high latitudes in Comp West which indicates stronger meridional transport in the lower stratosphere. The reason is twofold: wave propagation and different vertical velocity. Following Haynes and Shuckburgh (2000), within westerly QBO winds waves are able to propagate across the equator and break, causing mixing, on the summer side of the westerlies. This results in more meridional mixing across the subtropical transport barrier into the summer hemisphere, indicated by higher mass mixing ratios in Comp West in the summerly northern*

*hemisphere in July.*

Splitting the plots into January and July we now can show the different behaviour of the vertical velocity in the winter hemisphere under CE. We add for the vertical velocity: Both months show a decrease of the downward residual vertical velocity ($\omega^*$) poleward of $60^o$ in the winter hemisphere, the main downward branch of the BDC.

**2. The authors in section 6 compare a simulation with 10T(s)/yr carried out with a 90-level (with QBO) and 39-level (without QBO) versions of the model in order to further demonstrate effects of the QBO. However, at no point in the manuscript do the authors show or discuss that properties of the models critical to aerosol transport are the same between the 39 and 90-level versions (except the QBO). For example, is the mean Brewer-Dobson circulation the same between these models? Is the dissipation from planetary waves (that directly impacts mixing) the same? What is the strength of stratospheric DJF and JJA jets? Are the temperature anomalies due to injections the same in both models? Without these aspects of the model demonstrated to be the same (or very similar), it is not clear here whether we're looking at differences due to the QBO or differences due to other model differences. I suggest that similarities between the aspects of the models mentioned above are shown in the appendix.**

Thank you for mentioning this important point. We should have taken more care regarding this. The two versions of the model have been compared for mainly meteorological variables in Schmidt et al (2013) and related to transport by Bunzel et al (2013). Many of the requested plots can be studied in these publications. Thus, we have not added them to the appendix. Bunzel et al (2013) came to the following conclusion:

'Results of 50-yr sensitivity simulations for different climate states, performed with the state-of-the-art GCM ECHAM6 in different model configurations, have shown several similarities in the appearance of the BDC among the applied configurations of the model. The BDC pattern is qualitatively similar, independent of vertical resolution and vertical extent of the model. Even the relative contribution of resolved and unresolved waves to the driving of the total upward mass flux from the troposphere to the stratosphere is comparable. Increasing the vertical resolution in the high-top model, on the other hand, results in a slightly slower BDC. The total upward mass flux at 70 hPa is reduced by roughly 5%, and the mean age of stratospheric air increases by about 20% in the midlatitudes. We attribute the origin of this difference in age of air among the two high-top models primarily to the reduced numerical diffusion through the tropopause in the L95 model configuration.'

We added to the model description (Sect 3.1):

*Bunzel and Schmidt (2013) compared simulations with low and high vertical resolution (47 and 95 levels) version of ECHAM6. The Brewer-Dobson Circulation is qualitatively similar and independent of resolution. The high resolution shows 5% less vertical mass flux and 20% increase in age of air at mid-latitudes. Numerical diffusion is reduced when increasing the vertical resolution, resulting in lower vertical extent of the sulfate layer. Schmidt et al. (2013) show that differences between the atmospheric mean states and trends simulated with two different vertical resolutions of ECHAM are in general small except for the tropics where the QBO is a dominant feature. Thus, we presume that we can assign differences between responses to sulfate aerosol forcing simulated with different vertical resolutions mostly to the internally generated or not generated QBO and to different strength of vertical numerical diffusion.*

However, we cannot clearly differ between changes due the QBO or due to other impact of different vertical resolution in our results. We say in the beginning of Section 6: *Niemeier and Timmreck (2015) used ECHAM5-HAM in a 39-layer version that does not allow for an internally generated QBO, simulating instead constant equatorial easterly winds. This allows us to estimate an error in*

*the sulfate forcing made by using a dynamically too simple model with, additionally, stronger numerical diffusion in vertical direction which presents itself like an additional artificial up- and downdraft. However, through the distinction of QBO phases in Section 5 we can clearly attribute effects simulated in this paper to changes in the tropical circulation.* We also change the text later in Section 6, as described under 3) in this answer to reviewer.

**3. Interpretation of differences between simulation 10Tg60 and Geo10 I and 10Tg30 (Figures 7 and 8) is not clear and very confusing. There is no convincing explanation in the text to account for the differences in AOD between the 3 simulations shown in Figure 7. There is an overall increase in aerosol number density in the accumulation moden Geo10 as compared to 10Tg60 (Figure 8), however if the contour level is adjusted to show the maximum contour, it could be that the distribution with latitude of aerosols looks very similar to that as in 10Tg60, and the overall change is due to something other than QBO winds.**

We decide to now show the effective radius instead of the particle density, which is shifted to the supplementary material. The effective radius shows clearly an increase of the particle size. Particle size impacts the AOD, which is smaller with the same burden but larger particles.

We changed the text to: *A clear effect is the roughly 20% lower AOD in 10Tg60 compared to Geo10 outside of the tropics with similar burden values in this region. The AOD is a measure of turbidity and degradation of sunlight and depends, for sulfate, on the particle size. Small particles scatter more efficiently than larger particles. The effective radius increases from 0.4 $\mu$m in Geo10 to 0.45 $\mu$m in 10Tg60 (Fig. 9) and, consequently, the AOD decreases. The better representation of the stratospheric dynamics in the higher resolution simulations and the resulting stronger confinement of the particles in the tropics cause them to grow larger.*

**Secondly, if the westerly QBO is inhibiting transport out of the tropics, why doesn't this apply to 10Tg30? There are plenty of aerosols in the accumulation mode in the extratropics and the QBO is in an even stronger westerly phase.**

In simulation 10Tg30 SO2 is injected at a level between the lower and the upper branch of the BDC. The plot of particle number of the accumulation mode indicates a transport path in the upper BDC branch. However, concentration of these particles is small and plays only a minor role in the overall picture as shown in the plot of sulfate concentration in the supplementary material.

AOD and burden of 10Tg30 show a strong maximum in the tropics and also the ratio of tropical to extratropical burden values show less meridional transport than Geo10 and 10Tg60.

**Besides, if QBOW intensified vertical transport as mentioned earlier in the manuscript, why is the distribution of coarse mode particles in Figure 8 for 10Tg60 (bottom center panel) confined to a smaller vertical region as compared to that in Geo10 (left bottom panel). This plot alone implies higher vertical velocity at the equator in Geo10 as compared to 10Tg60. Again, this difference could be the consequence of different model dynamics, wave breaking and BD circulation due to differences in vertical resolution, and not to the QBO.**

It has not been our intention to say that differences between Geo10 and 10Tg60 are only related to the QBO, especially as there is no QBO in 10Tg60. We said in the text: *... to estimate an error in the sulfate forcing made by using a dynamically too simple model....* However, we try to describe this better in the text. We add in the model description and in Section 6 some sentences on the impact of numerical diffusion. Stronger numerical diffusion in vertical direction is the main reason for the broader vertical sulfate layer in Geo10. This is caused by the lower vertical resolution in Geo10.

We add in the model description: *Numerical vertical diffusion is reduced when increasing the vertical resolution, resulting in a lower vertical extend of the sulfate layer.*

We add in the text in Sect 6: *We performed simulations with an injection rate of 10 Tg(S) yr$^{-1}$ in order to enable a direct comparison to the results of (Niemeier and Timmreck, 2015). They*

*used ECHAM5-HAM in a 39-layer version that would not simulate an internally generated QBO, but instead constant equatorial easterly winds. This allows us to estimate an error in the sulfate forcing made by using a dynamically too simple model* *with, additionally, stronger numerical diffusion in vertical direction which present itself as an additional artificial up- and downdraft. However, through the distinction of QBO phases in Section 5 we can clearly attribute effects simulated, in this paper, to changes in the tropical circulation.*

**Authors state in line 360 that the number density in the coarse mode is lower in Geo10 then in 10Tg60. I don't see that from Figure 8 — that maybe true right at the equator in a very small region, but overall the number density in the coarse mode if anything is bigger in Geo10.**
We now show the effective radius in the paper. This indicates better the shift to larger particles. The number density is showing this shift in the changed ratio between accumulation to coarse mode particles.

4. **Interpretation of differences in the text between 10Tg60 and 10Tg30 are also not consistent with the figures. For example: Lines 350-351: The difference between 10Tg60 and 10Tg30 is primarily the injection altitude and not the tropical wind system From Figure 10Tg60 shows that likely the aerosol transport out of the tropics occurs via the lower branch of the BDC, where in 10Tg30, the aerosols are transported via mixing and the upper branch of the BDC (see also minor comment 1).**
Line 350 to 351 were a very general statement on the differences of burden and AOD in the extratropics. The first paragraph of Section 6.1 changed to:  *The comparison of simulation Geo10, the low resolution model version without internally generated QBO, and simulation 10Tg60 shows in tropics and sub-tropics the impact of the different vertical resolution. The imperfect representation of stratospheric dynamics with constant easterly winds in Geo10 cause a more permeable barrier (Punge et al., 2009) resulting in a stronger meridional transport.* *The ratio of tropical to extratropical mean burden is lower in Geo10 than in 10Tg60 (Tab. 1), indicating the stronger transport.* *The model version with lower vertical resolution tends to overestimate meridional transport, a tendency also seen in earlier volcano studies (Niemeier et al. (2009); Timmreck et al. (1999).*
We also discuss now the transport of accumulation mode particles in 10Tg30 in the upper branch of the BDC in Section 6.1 (see next paragraph) and, additionally, we add some lines to Section 2.1 (see minor comment 1).

**Lines 366- 367: 'Figure 8 indicates low meridional transport resulting in low particle number densities in extratropics in 10Tg30.' This is not at all consistent with top right — most panel in Figure 8: there are plenty of particles in the extratropics.**
It's true that the text was not very clear. We change the description of the transport at several places in the text. The mentioned paragraph changes to: *Increasing the injection height to 30 hPa, simulation 10Tg30, further intensifies the equatorial* *confinement, which is stronger at a height of 30 hPa than at a height of 60 hPa. Thus, injection in 10Tg30 results in a strong tropical maxima* *of burden and AOD, as discussed in Section 5.2. Small particles with little sedimentation are transported vertically in the tropical pipe and meridionally in the upper branch of the BDC while coarse mode particles are transported in the lower branch (Fig 2 supplementary material). In the extratropics the AOD is 30% to 50% lower compared to simulation 10Tg60.*  *Due to the stronger tropical confinement the particles grow to radii up to 0.75 $\mu$m, an increase of about 0.25 $\mu$m compared to*

*10Tg60. The reduction of AOD in the extratropics is strong enough to reduce the global mean AOD of 10Tg30 compared to 10Tg60 (Tab. 1).*

**5. In order to explain the differences between AOD in the 3 simulations shown in Figure 7, it would be helpful to plot effective radius of particles and surface area density, instead of what is currently in Figure 8.**

We add a figure of effective particle radius to the text and move the figure of particle number density to the supplementary materials.

**Minor Comments:**

**1. Line: 77-78: 'This quasi-horizontal mixing is the main transport branch for sulfate aerosols'. This statement highly depends on the location of injection of the aerosols. There are three main ways the aerosols can be transported out of the tropics: a) The deep branch of the BDC, the shallow branch, and horizontal mixing. Aerosols injected right above the tropical tropopause are mostly going to be transported with the shallow branch of the BDC, those injected several kilometers above the tropopause will likely be primarily transported with the upper branch of the BDC. Some will be transported horizontally by mixing. I suggest that a discussion of the different branches of the BDC is added and how the location of injection (30 hPa and 60 hPa discussed here) affect which branch of the BDC is the primary transport mechanism. Figure 1 of Bonisch et al. 2011 has an excellent graphic (Atmos. Chem. Phys., 11, 3937-3948, 2011 www.atmos-chem-phys.net/11/3937/2011/ doi:10.5194/acp-11-3937-2011)**

We change the text in Section 2.1: *Sharp gradients of potential vorticity at the edges of the surf zone act as a transport barrier: the polar vortex at high latitudes inhibits transport to the poles in winter months and the equatorial jets of the QBO form, at this edges, the subtropical transport barrier. This subtropical barrier results in the formation of a reservoir for chemical species in the lower tropical stratosphere (Trepte and Hitchman, 1992). The barrier is strongest in heights between 21 km to 28 km (50 to 15 hPa). The strength of the transport barrier depends on the phase of the QBO. These barriers can be seen as 'eddy-transport-barriers' (Mcintyre, 1995). They do not act as barriers to the zonally averaged BDC (Haynes and shuckburgh, 2000). As a consequence, the BDC has two horizontal transport branches, one below and one above the transport barrier. Transport of sulfate out of the tropics occurs mainly in the lower branch of the BDC but for small particles in high level injection scenarios also in the upper branch.*

**2. Line 92: An average period of the QBO is 28 (not 29) months.**

Done

**3. Figure 1: Why isn't the QBO included here for the 4 Tg 30 hPa injection? Please include it.**

Done

**4.Temperature anomalies in rightmost panel of Figure 2 clearly exceed the colorbar. Please change the colorbar so the maximum and minimum temperature anomalies are clear in all the panels. Please in the text also include the amplitude of maximum temperature anomalies for the simulations in Figure 2.**

Done

**5. Line 203-204: 'Positive anomaly does not extend to the pole ... because polar vortex blocks the transport' - it would be helpful to overplot the aerosol concentrations here to demonstrate this point clearly.**

Done.

**6. Why is there such a strong negative temperature anomaly near 5 hPa at the equator in the 8Tg30 hPa simulation? (Rightmost panel of Figure 2)**

In the upper stratosphere/lower mesosphere, a significant negative temperature anomaly in the equatorial region and a significant positive temperature anomaly over the extratropics. This has been observed after strong volcanic eruptions.

The diabatic heating of aerosols in the lower stratosphere causes an increase in the residual vertical wind velocity in the tropical pipe and consequently an adiabatic cooling, which dominated the temperature anomaly in the region above the aerosol heating. The residual vertical wind in 10Tg60 and 10Tg30 (Figure 1, supplementary material) is strongest in the region of the strongest vertical wind shear (15 hPa (10Tg60) and 5 hPa (10Tg30), respectively). In this region the adiabatic cooling is strongest (1 K/d and 2 K/d), which corresponds to the height of the temperature minima of both results.

We add in the text: *The temperature anomalies in the upper stratosphere, including the cooling above the heated aerosol layer in the tropics, are caused by the increase of the residual vertical wind and the related adiabatic heating anomalies (Toohey et al., 2014).*

**7. Figure 3: The color-scale is inappropriate. It is impossible to see what are the zonal wind velocities in the top panels as well as anomalies in the bottom panels. Both clearly exceed the color scale. Please correct.**

Done

**8. Figure 8: Again here, the colorbar needs to be adjusted that it is clear what the maximum contour is in the top leftmost panel.**

We change the figure to effective radius.

**9. Line 445-447: That is too strong of a conclusion! Injecting at 30 hPa and other location could be viable at other latitudes – only equatorial injections have been shown here, hence authors should not make such a sweeping conclusion.**

We change the text to: A conclusion from this results can be that injecting at high levels at the equator might be unfavorable for CE, not only because it is technically more demanding.

**10. Lines 457- 469: I'm not sure how this paragraph is relevant to the main point of the study. If the authors chose to keep it, please explain how you arrived at the injection estimates up to 2100 mentioned in line 460 and 462.**

We change the paragraph to:

In this study we calculate a smaller efficiency of sulfur injections than (Niemeier and Timmreck, 2015) obtained in model simulations with lower vertical resolution and, hence, less realistic tropical dynamics. Therefore we have to modify some of the conclusions drawn in (Niemeier and Timmreck, 2015). They estimated from their simulations an injection of 45 Tg(S) yr$^{-1}$ would counteract global greenhouse gas forcing of 6 W/m$^{-2}$. This amount would be necessary to keep the global mean temperature at 2020 level in 2100 while maintaining business as usual emissions. The decreased forcing efficiency simulated in this study would increase the injected amount to 70 Tg(S) yr$^{-1}$, injected at the equator, as lowest estimate to keep 2020 temperature level in 2100. Adapting a strategy of Laakso et al (2017), with injections following the zenith of the sun or injecting at 15 N and 15 S, may slightly reduce the injection rate. However, the spread in the forcing simulated by different model is large (Niemeier and Tilmes, 2017), as is the amount of injected sulfur necessary to generate a certain forcing. Estimates of lifting costs of sulfur into the stratosphere (e.g. Moriyama et al. (2016)) depend strongly on the efficiency of the injection.

**References**

Bunzel, F. and Schmidt, H.: The Brewer-Dobson Circulation in a Changing Climate: Impact of the Model Configuration, J. Atmos. Sci., 70, 1437–1455, doi:http://dx.doi.org/10.1175/JAS-D-12-0215.1, 2013.

Haynes, P. and Shuckburgh, E.: Effective diffusivity as a diagnostic of atmospheric transport. I-Stratosphere, Journal of geophysical research, 105, 22, 2000.

Mcintyre, M.: The Arctic and environmental change - The stratospheric polar vortex and sub-vortex : fluid dynamics and midlatitude ozone loss, Philosophical Transactions of the Royal Society of London A: Mathematical, Physical and Engineering Sciences, 352, 227–240, doi:10.1098/rsta.1995.0066, URL http://rsta.royalsocietypublishing.org/content/352/1699/227, 1995.

Moriyama, R., Sugiyama, M., Kurosawa, A., Masuda, K., Tsuzuki, K., and Ishimoto, Y.: The cost of stratospheric climate engineering revisited, Mitigation and Adaptation Strategies for Global Change, pp. 1–22, doi:10.1007/s11027-016-9723-y, URL http://dx.doi.org/10.1007/s11027-016-9723-y, 2016.

Niemeier, U. and Tilmes, S.: Sulfur injections for a cooler planet, Science, 357, 246–248, doi: 10.1126/science.aan3317, URL http://science.sciencemag.org/content/357/6348/246, 2017.

Niemeier, U. and Timmreck, C.: What is the limit of climate engineering by stratospheric injection of SO2?, Atmospheric Chemistry and Physics, 15, 9129–9141, doi:10.5194/acp-15-9129-2015, URL http://www.atmos-chem-phys.net/15/9129/2015/, 2015.

Niemeier, U., Timmreck, C., Graf, H.-F., Kinne, S., Rast, S., and Self, S.: Initial fate of fine ash and sulfur from large volcanic eruptions, Atmospheric Chemistry and Physics, 9, 9043–9057, URL http://www.atmos-chem-phys.net/9/9043/2009/, 2009.

Punge, H. J., Konopka, P., Giorgetta, M. A., and Müller, R.: Effects of the quasi-biennial oscillation on low-latitude transport in the stratosphere derived from trajectory calculations, J. Geophys. Res., 114, D03 102, doi:10.1029/2008JD010518, 2009.

Schmidt, H., Rast, S., Bunzel, F., Esch, M., Giorgetta, M., Kinne, S., Krismer, T., Stenchikov, G., Timmreck, C., Tomassini, L., and Walz, M.: Response of the middle atmosphere to anthropogenic and natural forcings in the CMIP5 simulations with the Max Planck Institute Earth system model, Journal of Advances in Modeling Earth Systems, 5, 98–116, doi:10.1002/jame.20014, URL http://dx.doi.org/10.1002/jame.20014, 2013.

Timmreck, C., Graf, H.-F., and Kirchner, I.: A one and half year interactive MA/ECHAM4 simulation of Mount Pinatubo Aerosol, J. Geophys. Res., 104, 9337–9360, doi:10.1029/1999JD900088, 1999.

Toohey, M., Krüger, K., Bittner, M., Timmreck, C., and Schmidt, H.: The impact of volcanic aerosol on the Northern Hemisphere stratospheric polar vortex: mechanisms and sensitivity to forcing structure, Atmospheric Chemistry and Physics, 14, 13 063–13 079, doi:10.5194/acp-14-13063-2014, URL https://www.atmos-chem-phys.net/14/13063/2014/, 2014.

Trepte, C. R. and Hitchman, M. H.: Tropical stratospheric circulation deduced from satellite aerosol data, Nature, 355, 626–628, 1992.

---

## Author Response (AR1)

[revised manuscript text omitted]

a transport barrier and the strong tropical westerly jet acts as a strong transport barrier. The stronger the jet, the stronger the barrier. Thus, both, stronger wave activity and stronger barrier, coexist.

For injections at 60 hPa we can identify a classical feedback loop: The stronger the injection, the
stronger the warming, which increases via the thermal wind imbalance the zonal wind velocity.This increases the tropical confinement of sulfate and results in an even higher heating. The stronger wind increases the transport barrier which keeps more sulfate within the tropics. This results in an even higher heating. Additionally, the results highlight the importance of quasi-horizontal mixing for the distribution of the surface aerosol. Transport in the BDC would be less sensitive to a stronger
subtropical transport barrier.

**6   Implications of changes in stratospheric sulfate transport for radiative forcing**

We have shown that radiative heating of the sulfate aerosols impacts the quasi-biennial oscillation by slowing or even shutting down the oscillation. In turn, the changed QBO impacts the meridional transport of the sulfate. What does this mean for the efficiency of CE? A good measure for the
efficiency is the TOA radiative forcing. This value is necessary to calculate the amount of $SO_2$ needed to counteract a certain amount of greenhouse forcing (Niemeier et al., 2013) and, therefore, determines the efficiency of a sulfur injection. It allows to estimate which forcing can be computed by a certain sulfur injection. In this study TOA forcing of sulfate is calculated as the difference between the net TOA flux with aerosols and a TOA flux without aerosols, which is obtained from
doubled radiative transfer calculations (see also Niemeier and Timmreck (2015)).

We perform simulations with an injection rate of $10\,\mathrm{Tg(S)yr^{-1}}$ in order to enable a direct comparison to simulation Geo10 of Niemeier and Timmreck (2015). They used ECHAM5-HAM in a 39-layer version that could not simulate an internally generated QBO, but instead constant equatorial easterly winds. This allows us to estimate an error in the sulfate forcing made by using a
dynamically too simple model with, additionally, stronger numerical diffusion in vertical direction due to the larger grid space, which present itself as an additional artificial up- and downdraft. However, through the distinction of QBO phases in Section 5 we can clearly attribute effects simulated in this paper to changes in the tropical circulation.

Mean wind velocities in the westerly jet increase by 5% (10Tg60) and 10% (10Tg30), respectively,

[revised manuscript text omitted]

with ECHAM5-HAM even $10\,\mathrm{Tg(S)yr^{-1}}$ would not be enough, $12\,\mathrm{Tg(S)yr^{-1}}$ might be necessary. Thus, a factor of 4 more planes and flights would be required than in the study of Moriyama et al.

(2016).

Finally, it needs to be stated that the simulated impact of stratospheric sulfate heating on the QBO is only a model result which cannot be evaluated in reality. However our simulations further show that the efficiency of sulfur injections may depend crucially on the jet structure in the tropical stratosphere, which itself will be influenced strongly by the injections. Our simulations show that the dynamical effects vary strongly even in different configurations of the same model. To reduce this uncertainty a better understanding of tropical dynamics and model simulations without the necessity of gravity wave parameterizations, i.e. with horizontal resolutions at least one order of magnitude higher than used here, may be necessary. As for many questions related to CE, certainty of response would require the full implementation of CE. Observations of a prolonged westerly phase of the

QBO (Labitzke, 1994) are weak due to the short lifetime of volcanic sulfate in the stratosphere. It would be nice to confirm the effect of sulfate aerosols on the QBO in observations after volcanic eruptions, but this is difficult due to the small number of well observed large tropical eruptions, the short lifetime of volcanic aerosols, and the internal variability of the QBO.

*Acknowledgements.* We thank two anonymous reviewers for their helpful comments, Simone Tilmes and Yaga

[revised manuscript text omitted]

---

## Author Response (AR2)

**Answers to the 2nd review on the ACPD paper (acp-2017-470)**
Changing transport processes in the stratosphere by radiative heating of sulfate aerosols

Ulrike Niemeier and Hauke Schmidt
Max Planck Institute for Meteorology, Bundesstr. 53, 20146 Hamburg, Germany

We thank the reviewer for the helpful comments. We followed the advices, added citations to the question on numeric diffusion and added significances to the right plot in Fig. 4.
Citations of the text are written in *italic* and changed or new text is highlighted in blue.

**1 Line 74: The phrase 'related to' is not the best choice of words here. Maybe use 'consists of'?**
We followed the reviewers advice.

**2 Line 159/160: 'Numerical diffusion is reduced.' Is that shown somewhere? Can a reference be added?**
We added cititions for an applied and theroretical appreach: (see Land et al. (2002) for an applied and Quarteroni et al. (2010) for an theoretical approach). The impact is also described in Bunzel and Schmidt (2013).

**3 Line 213: 'caused by the absorption of LW' is it only LW or also SW that is absorbed? Previous work by Ferraro indicates that it's both.**
Yes. Sulfate is absorbing in the near infrared and infrared, as we mention in the introduction. The near infrared is part of the SW band in the model. We deleted the LW in the text.

**4 Rightmost panels of Figure 4 could use significance testing.**
Done

**5. Line 377: From the Figure in supplement it looks like the differences in temperature anomalies are greater than 1 K (2 more contours which seem like 1K contours); Similarly, for vertical velocity it's a bit hard to tell with current contours, but there seems to stronger vertical wind at 30 hPa in 10Tg60.**
We changed the text to:
*Zonal wind and temperature profiles are similar to the 8Tg results (see also Figure 1 in the supplementary material), while the model resolution without internally generated QBO used in Geo10 simulates easterly winds in the tropics and subtropics. The temperature anomaly is about 1 to 2 K higher in 10Tg60 than in Geo10, but the residual vertical wind velocity is similar at the height of 50 hPa, the level of the concentration maximum (Fig. 4) and slighly higher in 10Tg30 above 35 hPa, the region of highest mass mixing rations (Figure 2 in the supplementary material).*

**References**

Bunzel, F. and Schmidt, H.: The Brewer-Dobson Circulation in a Changing Climate: Impact of the Model Configuration, J. Atmos. Sci., 70, 1437–1455, doi:http://dx.doi.org/10.1175/JAS-D-12-0215.1, 2013.

Land, C., Feichter, J., and Sausen, R.: Impact of vertical resolution on the transport of passive tracers in the ECHAM4 model, Tellus B, 54, 344–360, doi:10.1034/j.1600-0889.2002.201367.x, 2002.

Quarteroni, A., Sacco, R., and Saleri, F.: Numerical Mathematics (Texts in Applied Mathematics) 2nd Edition, Springer, Berlin Heidelberg, doi:10.1007/b98885, 2010.